# Testicular macrophages are recruited during a narrow fetal time window and promote organ-specific developmental functions

Xiaowei Gu [1], Anna Heinrich[1], Shu-Yun Li[1] & Tony DeFalco [1,2] ✉

A growing body of evidence demonstrates that fetal-derived tissue-resident macrophages have developmental functions. It has been proposed that macrophages promote testicular functions, but which macrophage populations are involved is unclear. Previous studies showed that macrophages play critical roles in fetal testis morphogenesis and described two adult testicular macrophage populations, interstitial and peritubular. There has been debate regarding the hematopoietic origins of testicular macrophages and whether distinct macrophage populations promote specific testicular functions. Here our hematopoietic lineage-tracing studies in mice show that yolk-sac-derived macrophages comprise the earliest testicular macrophages, while fetal hematopoietic stem cells (HSCs) generate monocytes that colonize the gonad during a narrow time window in a Sertoli-cell-dependent manner and differentiate into adult testicular macrophages. Finally, we show that yolk-sac-derived versus HSC-derived macrophages have distinct functions during testis morphogenesis, while interstitial macrophages specifically promote adult Leydig cell steroidogenesis. Our findings provide insight into testicular macrophage origins and their tissue-specific roles.

Macrophages are ubiquitous throughout the body and play tissue-specific roles, often with unique functions critical for organ development and homeostasis that are dramatically distinct from their traditionally appreciated phagocytic and antigen-presenting roles[1,2]. Recent research in the field has focused on the hematopoietic origins of macrophages during embryogenesis and their potentially diverse repertoire of cellular functions during fetal development[3,4]. A major open question is whether hematopoietic origins dictate the capability of macrophages to perform tissue-specific activities, or if the cellular microenvironment in which macrophages reside is the major driving factor underlying tissue-resident macrophage phenotypes and activities.

Using specific labeling and targeting tools in the mouse model system, numerous reports have shed light onto new models of hematopoiesis that diverge from a classical model in which all macrophages were assumed to differentiate from circulating monocyte progenitors originating from adult bone marrow hematopoietic stem cells (HSCs) in what is termed definitive hematopoiesis[5]. Studies over the past 15 years have demonstrated that, in contrast to the previous dogma centered around adult definitive hematopoiesis, embryonic hematopoiesis is a major contributor to tissue-resident macrophages within adult organs[6–12]. Furthermore, a number of organs, such as the brain, maintain their tissue-resident macrophage populations (microglia) independently of bone marrow-derived HSCs, while other organs, such as the lung and gut, exhibit a heavy reliance on fetal-liver-derived monocytes and bone marrow–HSC-derived monocytes, respectively, to maintain homeostasis of their tissue-resident macrophages[4]. It is clear that each organ has unique dynamics and kinetics of macrophage colonization and homeostasis during development; however, the underlying mechanisms that drive these organ-specific phenotypes are unclear.

[1]Reproductive Sciences Center, Division of Developmental Biology, Cincinnati Children's Hospital Medical Center, Cincinnati, OH 45229, USA. [2]Department of Pediatrics, University of Cincinnati College of Medicine, Cincinnati, OH 45267, USA. ✉e-mail: tony.defalco@cchmc.org

Mammalian hematopoiesis occurs in several waves to give rise to cells that seed organs and differentiate into tissue-resident macrophages[3,4,13]. The first wave, called primitive hematopoiesis, takes place in the embryonic yolk-sac (YS) blood islands starting at embryonic day (E) 7.0 in mice and gives rise to early erythromyeloid precursors (EMPs) without the capacity of monocyte output. Between E8.0 and E8.5, a second wave of hematopoiesis from YS hemogenic endothelium, sometimes referred to as the transient wave before fetal HSCs emerge and differentiate, gives rise to a class of progenitors called late EMPs that migrate to the fetal liver and expand in number to differentiate into monocytes and other hematopoietic lineages[3,4]. Soon thereafter, within the hemogenic endothelium of the embryo proper, immature HSCs arise in the para-aortic splanchnopleura region and give rise to fetal HSCs at E10.5 within the aorta-gonad–mesonephros region (AGM). These HSCs then colonize the fetal liver to initiate steady-state definitive hematopoiesis in the embryo; starting at E12.5, the fetal liver is the main site of hematopoiesis where myeloid and lymphoid cells of the nascent embryonic immune system will be generated. By the end of embryogenesis, fetal-liver HSCs will seed the fetal bone marrow, where these fetal HSCs will give rise to adult bone marrow HSCs[4,13].

The gonad, and the testis in particular, is a unique immune environment where inflammatory responses are generally suppressed to protect gametes from attack by the immune system and to ensure continued fertility[14–16]. A major component of this specialized immune environment is the testicular macrophage, which is a central contributor to testicular immune function[17–19] and comprises a significant proportion (up to 25% of cells) of the interstitial compartment of the testis[20]. We have previously shown that macrophages are present in the early undifferentiated fetal gonad and YS hematopoietic precursors give rise to macrophages in the nascent fetal testis[21], and other recent reports have also examined the origins of testicular immune cells, implicating monocytes and other myeloid cell types[22–24]. In terms of function, previous reports demonstrated that fetal testicular macrophages are required for testis morphogenesis during initial gonadogenesis, likely through modulation of vascular development that is critical for the formation of testicular architecture[21,25].

At least two major adult testicular macrophage populations have been described: CSF1R+CD206+MHCII− interstitial macrophages, which are embedded amongst Leydig cells and blood vessels in the interstitial parenchyma; and CSF1R−CD206−MHCII+ peritubular macrophages, which are exclusively localized to the surface of seminiferous tubules[23,26]. There has been some debate regarding the hematopoietic origin of adult testicular macrophage populations, with one study proposing a circulating, bone marrow–HSC-derived origin for peritubular macrophages, and others proposing a fetal–monocyte-derived origin for these cells[22–24]. Testicular macrophages have long been implicated in regulating Leydig cell steroidogenesis[27–29], while peritubular macrophages have been proposed to promote spermatogonial differentiation[26]; however, specific functions for distinct testicular macrophage populations have not been functionally investigated.

In this work, we investigate the hematopoietic origins and functions of testicular macrophages. Using a wide array of lineage-tracing tools in mice, we find that YS-derived macrophages colonize the early nascent testis but make a negligible contribution to adult testicular macrophages; in contrast, monocytes originating from AGM-derived HSCs recruited during a specific embryonic time window are the major hematopoietic precursor to adult testicular macrophage populations. Our analyses demonstrate that Sertoli cells are required to recruit monocytes into the fetal testis, whereas germ cells are dispensable for recruitment, differentiation, and localization of adult interstitial and peritubular macrophages. We specifically ablate YS fetal macrophages and find that testicular cord morphogenesis is disrupted and ectopic monocytes are observed in the testis; our data suggest that distorted myeloid cell ratios specifically impact testis cord morphogenesis.

Finally, we use in vitro assays to demonstrate that testicular CD206+ interstitial macrophages specifically promote Leydig cell proliferation and steroidogenesis, providing detailed insights into previous long-standing questions in the field. Our findings highlight the complexity of testicular macrophage origins and functions, and provide perspective into mechanisms that potentially underlie gonadal developmental defects and infertility.

## Results

### *Csf1r*+ fetal definitive progenitors give rise to adult testicular macrophages

Previous reports have shown that induction with 4-hydroxytamoxifen (4-OHT) at E8.5 in the *Csf1r*-creER mouse model can specifically target YS-derived EMPs[30,31]. To identify whether YS-derived EMPs or subsequent *Csf1r*+ definitive progenitors contribute to fetal and postnatal testicular macrophages, we crossed *Csf1r*-creER mice with *Rosa*-Tomato reporter mice and administered a single injection of 4-OHT at E8.5, E10.5 or E12.5 (Fig. 1a). To address the general specificity of cell targeting assays in this study, we examined several aspects of the creER/*Rosa*-Tomato labeling system, including effects of 4-OHT on cell cycle activity and apoptosis in fetal testes (Supplementary Fig. 1), baseline reporter allele activity (Supplementary Fig. 2), and acute versus long-term impacts of 4-OHT administration on fetal immune cell number (Supplementary Figs. 1 and 2). We did not find any spurious or non-specific reporter activity in the absence of 4-OHT, and there was no significant effect on any of the above parameters, except a slight long-term increase in testicular non-macrophage immune cells (CD45+F4/80− cells, which are likely monocytes) after 4-OHT treatment (Supplementary Fig. 2), thus indicating that our experiments are highly specific and represent normal hematopoietic and testicular development.

When pulsed with 4-OHT at E8.5, as a positive control, we examined microglia, which are derived exclusively from YS progenitors[7,11,30], in E18.5 and postnatal brains. As expected, there was extensive Tomato labeling (70–80%) of F4/80+ microglia throughout the brain (Supplementary Fig. 3a, b, m), whereas only about 15% of F4/80+ macrophages expressed Tomato in the E18.5 testis (Fig. 1b, o). Even fewer (less than 5%) CSF1R+MHCII− or CD206+MHCII− interstitial macrophages and CSF1R−MHCII− or CD206−MHCII+ peritubular macrophages expressed Tomato at postnatal day 30 (P30) and P90 (Fig. 1c, d, q, r), suggesting that YS-derived EMPs had a negligible contribution to adult testicular macrophages. We also could induce labeling of microglia as efficiently with an E10.5 pulse of 4-OHT (>80%) as with an E8.5 pulse, with no dilution from fetal to adult stages, but E12.5 induction only labeled about 40% of microglia (Supplementary Fig. 3c–f, m), confirming an earlier report that YS-macrophage progenitors travel to tissues within a restricted time window[32]. However, while Tomato-labeled liver macrophages (Kupffer cells) were diluted from fetal to adult stages when induced at E8.5 or E10.5 (Supplementary Fig. 3g–j, n), a large number of Kupffer cells were stably labeled when induced at E12.5 (Supplementary Fig. 3k, l, n). These findings indicate that EMP-derived Kupffer cells are partially replaced by *Csf1r*+ definitive progenitor-derived macrophages, supporting a previous study proposing Kupffer cells come from two sources: YS-derived EMPs and fetal immature HSCs[7,33]. These data suggest that *Csf1r*+ macrophage progenitors produced at E12.5 are no longer strictly derived from the YS.

When pulsed at E10.5 and E12.5, the majority of F4/80+ macrophages in E14.5, E16.5, and E18.5 testes expressed Tomato (Fig. 1e–j, o). In addition, high numbers of Tomato-labeled interstitial and peritubular macrophages in P30 and P90 testes were detected when induced at E10.5 or E12.5, and their frequency had the most substantial increase in E12.5-induced P30 and P90 testes (Fig. 1k–n, q, r).

Of note, we found that induction at E10.5 yielded some Tomato+F4/80− cells in testes starting from E16.5 (Fig. 1e–g, p), whereby induction at E12.5 had more such cells emerging in late fetal

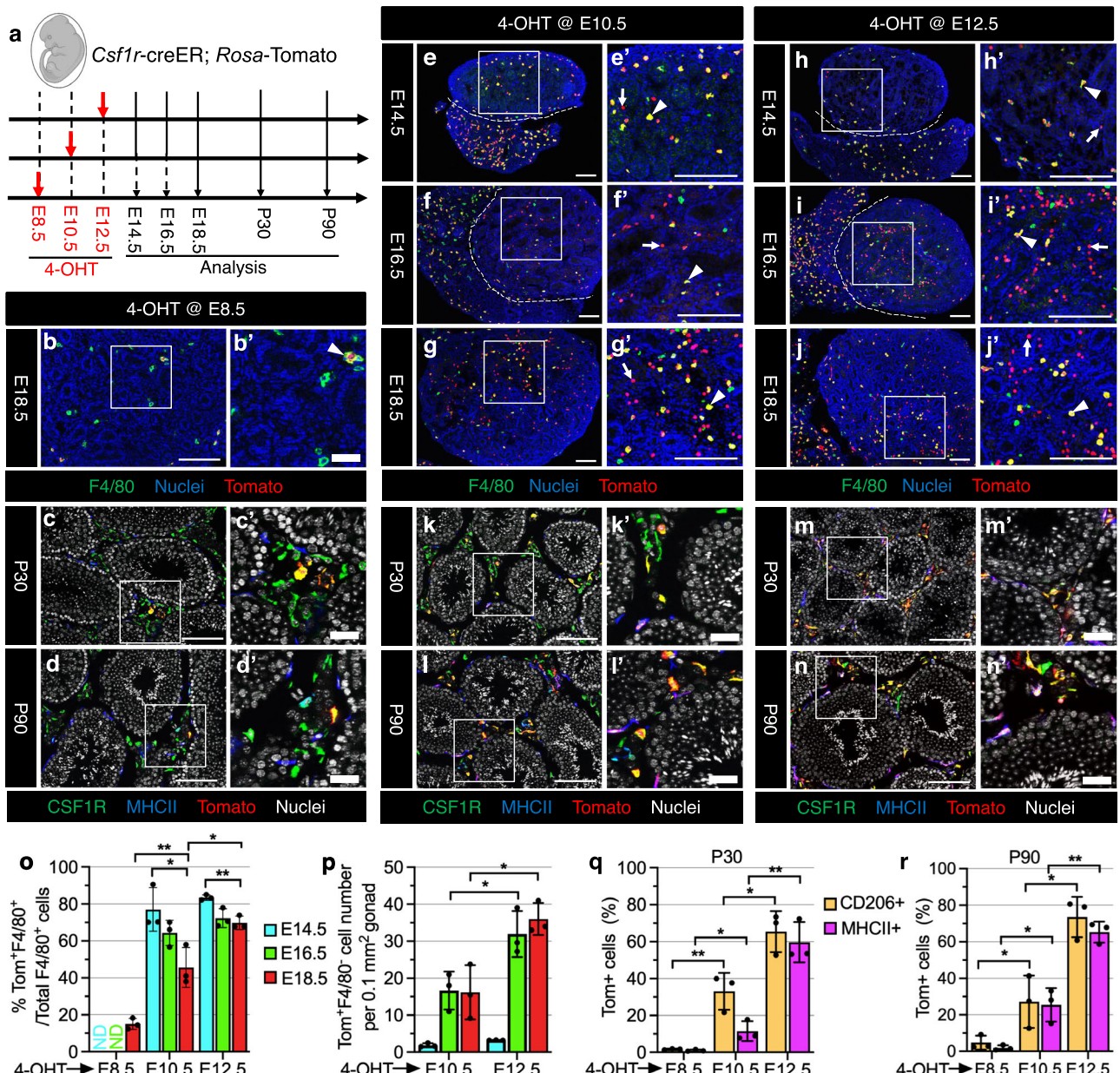

**Fig. 1 | Csf1r-expressing fetal definitive progenitors give rise to adult testicular macrophages. a** Strategy for 4-hydroxytamoxifen-induced (4-OHT) lineage-tracing and harvesting of testes from *Csf1r*-creER; *Rosa*-Tomato embryos and juvenile/adult mice. The embryo image was created with BioRender.com software (BioRender.com). **b–n** Representative images (*n* = 3) of testes at various stages from *Csf1r*-creER; *Rosa*-Tomato mice exposed to 4-OHT at E8.5 (**b–d**), E10.5 (**e–g**, **k**, **l**), or E12.5 (**h–j**, **m**, **n**). In all figures throughout this study, prime figures (e.g., **a′** relative to **a**) are higher-magnification images of the boxed regions in the image to their left. Dashed lines indicate the gonad−mesonephros boundary. Arrowheads denote

Tomato-expressing F4/80⁺ macrophages and arrows denote Tomato-expressing F4/80-negative cells. Thin scale bar, 100 μm; thick scale bar, 25 μm. **o−r** Graphs showing quantification (*n* = 3 independent gonads) of percent Tomato-expressing F4/80⁺ macrophages at E14.5, E16.5, or E18.5 (**o**), number of Tomato-expressing F4/80⁻ cells per unit area at E14.5, E16.5, or E18.5 (**p**), and percent Tomato-expressing interstitial (CD206⁺) and peritubular (MHCII⁺) macrophages at P30 (**q**) or P90 (**r**) in *Csf1r*-creER; *Rosa*-Tomato testes induced with 4-OHT at various embryonic stages. Data are shown as mean +/− SD. *$P < 0.05$; **$P < 0.01$ (two-tailed Student's *t* test). Exact *P* values are provided in the Source Data file.

testes (Fig. 1h–j, p). When induced at E10.5 or E12.5, all Tomato⁺ cells in E18.5 liver and testes were positive for CD45 (a pan-leukocyte marker) (Supplementary Fig. 4a–d), but some KIT⁺ (also called C-KIT or CD117; a HSC marker) cells in the liver expressed Tomato (Supplementary Fig. 4e, f), suggesting *Csf1r*-creER-driven Tomato-labeled progenitors may be the progeny of fetal HSCs. In addition, some GR1⁺ cells, primarily representing granulocytes and monocytes, were also labeled in E18.5 testes and liver (Supplementary Fig. 4g–j). Flow cytometry of E18.5 fetal liver and blood demonstrated that KIT⁺ HSCs, Ly6C⁺ monocytes, Ly6G⁺ neutrophils, and F4/80⁺ macrophages were all extensively labeled by Tomato, indicating that HSCs were targeted in

this assay; however, the level of labeling in differentiated immune cell types was higher than in KIT⁺ cells (Supplementary Fig. 5).

## AGM-derived HSCs contribute to adult testis macrophages during a specific recruitment window

Although the *Csf1r*-inducible fate-mapping model targets fetal HSC-derived multipotent progenitors, *Csf1r* is also expressed in mature macrophages[34,35], complicating the interpretation of the results. Given that KIT is widely expressed in fetal and adult HSCs as well as in early YS and fetal-liver progenitors, but not in mature hematopoietic cells[36], we used the *Kit*^creER mouse strain to trace hematopoietic progenitors and

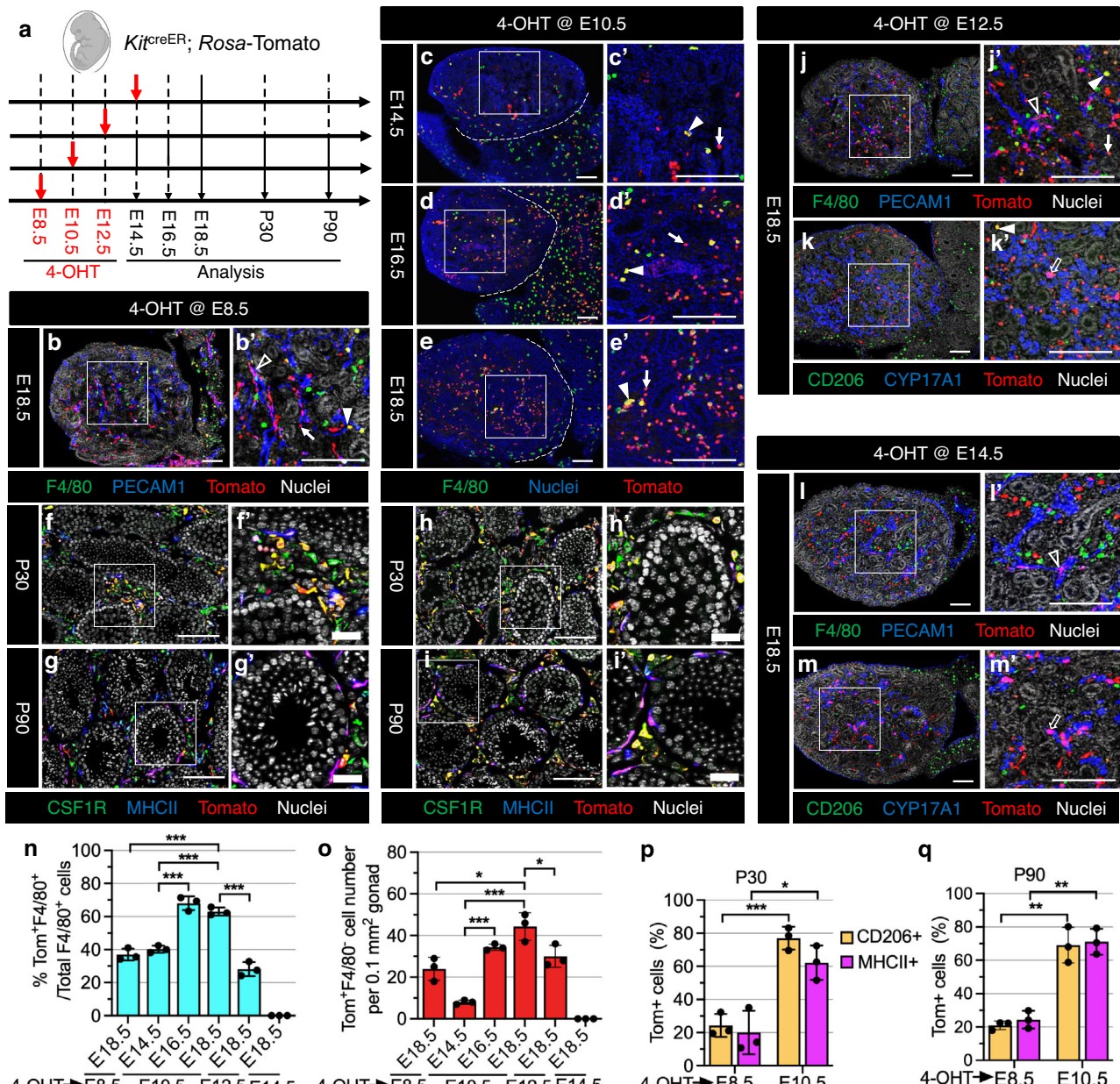

**Fig. 2 | Contribution of AGM-derived HSCs to adult testicular macrophages occurs within a specific recruitment window during fetal development.**
**a** Strategy for 4-OHT-induced lineage-tracing and harvesting of testes from $Kit^{creER}$; $Rosa$-Tomato embryos and juvenile/adult mice. Embryo image was created with BioRender.com software (BioRender.com). **b-m** Representative images ($n = 3$ independent gonads) of testes at various stages from $Kit^{creER}$; $Rosa$-Tomato mice exposed to 4-OHT at E8.5 (**b, f, g**), E10.5 (**c-e, h, i**), E12.5 (**j, k**), or E14.5 (**l, m**). Dashed lines indicate gonad–mesonephros boundary. Black arrowheads denote Tomato-expressing PECAM1⁺ endothelial cells, white arrowheads denote Tomato-expressing F4/80⁺ macrophages, black arrows denote Tomato-expressing

CYP17A1⁺ Leydig cells, and white arrows denote Tomato-expressing F4/80-negative cells. Thin scale bar, 100 μm; thick scale bar, 25 μm. **n–q** Graphs showing quantification ($n = 3$ independent gonads) of percent Tomato-expressing F4/80⁺ macrophages at E14.5, E16.5, or E18.5 (**n**), number of Tomato-expressing F4/80⁻ cells per unit area at E14.5, E16.5, or E18.5 (**o**), and percent Tomato-expressing interstitial (CD206⁺) and peritubular (MHCII⁺) macrophages at P30 (**p**) and P90 (**q**) in $Kit^{creER}$; $Rosa$-Tomato testes induced with 4-OHT at various embryonic stages. Data are shown as mean +/− SD. *$P < 0.05$; **$P < 0.01$; ***$P < 0.001$ (two-tailed Student's $t$ test). Exact $P$ values are provided in the Source Data file.

HSCs at different time points during gestation (E8.5, E10.5, E12.5, E14.5) in $Kit^{creER}$; $Rosa$-Tomato embryos (Fig. 2a).

When pulsed at E8.5, we detected Tomato⁺F4/80⁺ testicular macrophages and Tomato⁺F4/80⁻ cells in E18.5 testes (Fig. 2b, n, o). In addition, microglia in E18.5 brains were labeled by Tomato (Supplementary Fig. 6a, c), which is consistent with a previous study showing the labeling of AGM-derived multipotent progenitors (MPPs)[11] with an E8.5 pulse, suggesting that induction at E8.5 targets YS-derived EMPs and AGM-derived HSCs.

When pulsed at E10.5, we observed no Tomato-labeled microglia in E18.5 brains (Supplementary Fig. 6b, c), while 25–30% of macrophages were labeled by Tomato in E18.5 and P30 liver, as well as extensive labeling of B220⁺ B cells and CD4⁺ T cells in the postnatal liver and spleen (Supplementary Fig. 6e–l), suggesting that induction at E10.5 labels AGM-derived HSCs that give rise to all hematopoietic cell lineages. We detected a small number of Tomato⁺ cells in E14.5 testes when induced at E10.5 (Fig. 2c), but there was a considerable increase in the proportion of Tomato⁺F4/80⁺ macrophages and in the

number of Tomato⁺F4/80⁻ cells from E16.5 to E18.5 when compared to induction at E8.5 (Fig. 2d, e, n, o). In E18.5 testes, all Tomato-expressing cells were CD45⁺ and divided into two populations, which were CD45⁺IBA1⁺ (IBA1 is a microglial/macrophage marker) and CD45⁺IBA1⁻ (Supplementary Fig. 6d), suggesting that CD45⁺Tomato⁺F4/80⁻IBA1⁻ cells in E16.5 or E18.5 testes likely represent monocytes, although they could be other immune cells such as neutrophils or other granulocytes. In addition, induction at E10.5 produced a higher number of Tomato⁺ interstitial and peritubular macrophages in P30 and P90 testes compared to induction at E8.5 (Fig. 2f–i, p, q). This may be due to the incomplete labeling of fetal HSCs when pulsed at E8.5. These results suggest that AGM-derived HSCs give rise to the vast majority of adult testicular interstitial and peritubular macrophages.

To further examine whether fetal-liver HSCs contribute to testicular macrophages, we induced Tomato expression in *Kit*^creER; *Rosa*-Tomato embryos at E12.5 or E14.5 when the fetal liver has become the dominant site of hematopoiesis[13]. In E18.5 testes that were induced at E12.5, there was a similar labeling efficiency of F4/80⁺ macrophages and a comparable Tomato⁺F4/80⁻ cell number as compared to induction at E8.5 (Fig. 2j, n, o). However, induction at E14.5 showed no Tomato-labeled testicular macrophages (Fig. 2l, n, o), suggesting that the contribution of fetal HSCs to testicular macrophages has a limited recruitment window. We also detected labeling of CYP17A1⁺ fetal Leydig cells (Fig. 2k, m), which is linked to KIT expression in fetal Leydig cells[37,38], as well as some PECAM1⁺ endothelial cells (Fig. 2b, j, l), supporting the previous observation that YS-derived EMPs contribute endothelial cells to various tissue blood vessels[31].

To address whether postnatal and adult HSCs contribute to testicular macrophages, we exposed *Kit*^creER; *Rosa*-Tomato pups and adult mice to tamoxifen (TAM) at postnatal stages P4-P5 and P60-P62, respectively (Supplementary Fig. 7). When injected at P4 and P5, we found that no testicular macrophages were labeled (Supplementary Fig. 7a–c), whereas endothelial cells (PECAM1⁺), spermatogonia (KIT⁺) and Leydig cells (KIT⁺, CYP11A1⁺, or NR5A1⁺) expressed Tomato in P7, P30 and P60 testes (Supplementary Fig. 7d–i). Consistent with induction at the neonatal stage, in adult-induced testes 60 days post-induction (P120) we also observed no Tomato-labeled interstitial or peritubular macrophages (Supplementary Fig. 7j), but instead observed extensive labeling of Leydig cells (Supplementary Fig. 7k, l). Intestinal macrophages were extensively labeled by Tomato in P30 and P60, but not P7, *Kit*^creER; *Rosa*-Tomato intestines, when pulsed at P4 and P5, as well as in P120 *Kit*^creER; *Rosa*-Tomato intestines when pulsed at P60-P62, consistent with previous reports that adult intestinal macrophages have a bone marrow origin[39,40] (Supplementary Fig. 8); these data demonstrate that bone marrow HSCs are effectively targeted in *Kit*^creER; *Rosa*-Tomato mice and that bone marrow-derived HSCs after birth make no contribution to testicular macrophages.

### *Flt3*⁺ fetal HSC-derived multipotent progenitors give rise to testis macrophages

To further confirm the contribution of HSCs to testicular macrophages, we created *Flt3*-cre; *Rosa*-Tomato mice, in which Cre activity is mainly activated in late fetal or adult HSC-derived MPPs, but not in YS-derived EMPs and mature macrophages[7,41,42]. We detected few Tomato⁺ cells in E14.5 testes (Fig. 3a); however, there were numerous Tomato⁺ cells in E14.5 liver where extremely few of them were F4/80⁺ macrophages or KIT⁺CD45⁻ HSCs, but instead were either KIT⁺CD45⁺ or KIT⁻CD45⁺ progenitors (Supplementary Fig. 9a, b), indicating that fetal-liver hematopoietic progenitors do not migrate towards the testis at this stage. Subsequently, from E16.5 to P7, *Flt3*-cre; *Rosa*-Tomato testes

exhibited extensive Tomato⁺ cells with a gradual increase in the percentage of F4/80⁺ testicular macrophages expressing Tomato in those stages as compared to E14.5 (Fig. 3b–d, i). As negative controls, in E18.5 liver and brain, less than 10% of macrophages expressed Tomato (Supplementary Fig. 9c, d, g), consistent with a previous study[7], suggesting that liver and brain macrophages have reached a steady-state equilibrium at early fetal stages. However, Tomato-expressing cells were CD45⁺KIT⁺ or CD45⁺KIT⁻ fetal HSC-derived progenitors in E18.5 liver and CD45⁺ myeloid cells in E18.5 testes (Supplementary Fig. 9e, f). In addition, the number of Tomato⁺F4/80⁻ cells during late fetal testis development increased and plateaued in E18.5 testes, but then declined in P7 testes (Fig. 3b–d, j), implying that Tomato⁺F4/80⁻ cells in fetal testes may be the precursors of F4/80⁺ macrophages and differentiate into mature macrophages after birth. Indeed, in P30 and P90 testes, we observed an extremely high labeling efficiency of interstitial and peritubular testicular macrophages (Fig. 3e–h, k). These results suggest that fetal HSC-derived *Flt3*⁺ progenitors give rise to adult testicular macrophages.

### Fetal testicular monocytes gradually differentiate into testicular macrophages after birth

We next attempted to establish whether fetal monocytes contribute to adult testicular macrophages, using the *Cx3cr1*^creER mouse model widely used in the fate-mapping of myeloid cells, specifically monocytes and macrophages. We generated *Cx3cr1*^creER; *Rosa*-Tomato mice, which were induced with 4-OHT at E12.5, E18.5, and with TAM at P4-P5 (Fig. 4a).

When pulsed at E12.5, we observed that almost 70% of F4/80⁺ macrophages were labeled with Tomato at E18.5 (Fig. 4b, k), but few F4/80-negative cells were labeled, which resulted in a low percentage of Tomato labeling among total CD45⁺ cells (Fig. 4l), suggesting that *Cx3cr1*-creER activity in early fetal-liver-derived monocytes is limited. Flow cytometry analyses of fetal liver and blood indicated that there was virtually no labeling of KIT⁺ cells, Ly6G⁺ neutrophils, or Ly6C⁺ monocytes, indicating that *Cxc3r1*-creER activity is restricted to the monocyte/macrophage lineage and does not label other peripheral blood cell types outside the immediate 4-OHT treatment window or HSCs (Supplementary Fig. 10).

In P30 and P90 testes, there were few Tomato⁺ interstitial and peritubular macrophages (Fig. 4c, d, m, n), suggesting that E12.5-induced testicular macrophages are severely diluted after birth in the adult. Given the presence of monocytes in E18.5 testes, we speculated that monocytes that colonize the gonad during the late fetal and perinatal period contribute to adult testicular macrophages. To test this hypothesis, we exposed *Cx3cr1*^creER; *Rosa*-Tomato embryos to 4-OHT at E18.5. As expected, induction at E18.5 had relatively higher labeling of interstitial and peritubular macrophages in P30 and P90 testes as compared to induction at E12.5 (Fig. 4e, f, m, n), but also had a reduction in the percentage of Tomato-labeled interstitial macrophages upon aging from P30 to P90 (Fig. 4m, n), indicating an incomplete labeling of monocytes in E18.5 testes. When injected at P4 and P5, we found that more than 90% of F4/80⁺ macrophages expressed Tomato in P7 testes (Fig. 4g, k). In addition, there was a significant increase in the percentage of Tomato-labeled testicular interstitial and peritubular macrophages at P30 and P90 (Fig. 4i, j, m, n), with interstitial macrophage labeling remaining steady upon aging (Fig. 4m, n). This may be due to the gradual increase of monocyte labeling from E12.5 to neonatal stages, with the gradual upregulation of CX3CR1 expression in fetal monocytes when they migrate from the fetal liver into their target tissue[8]. We also found a large number of unlabeled CD45⁺IBA1⁻ cells in P7 testes (Fig. 4h, l), which were likely responsible for the low labeling efficiency of peritubular macrophages, which

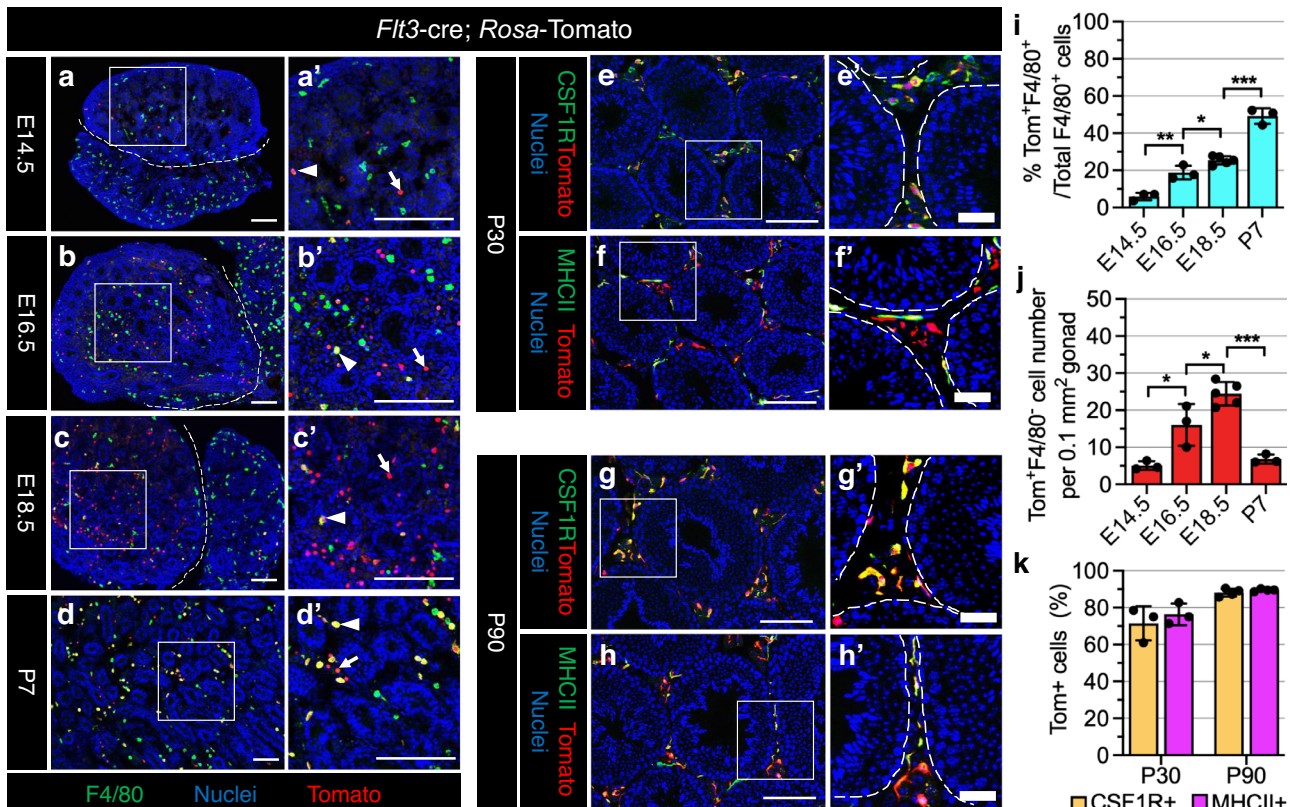

**Fig. 3 | Flt3-expressing fetal HSC-derived multipotent progenitors give rise to adult testicular macrophages. a–h** Representative images (*n* = 3 independent gonads) of *Flt3*-cre; *Rosa*-Tomato testes at E14.5 (**a**), E16.5 (**b**), E18.5 (**c**), P7 (**d**), P30 (**e**, **f**), and P90 (**g**, **h**). Arrowheads denote Tomato-expressing F4/80⁺ macrophages and arrows denote Tomato-expressing F4/80-negative cells. Thin scale bar, 100 μm; thick scale bar, 25 μm. **i–k** Graphs showing quantification of percent Tomato-expressing F4/80⁺ macrophages (**i**), number of Tomato-expressing F4/80-negative cells per unit area at E14.5 (*n* = 3 independent gonads), E16.5 (*n* = 3), E18.5 (*n* = 5) or P7 (*n* = 3) (**j**), and percent Tomato-expressing interstitial (CSF1R⁺) and peritubular (MHCII⁺) macrophages at P30 (*n* = 3) and P90 (*n* = 4) (**k**) in *Flt3*-cre; *Rosa*-Tomato testes. Data are shown as mean +/− SD. *P < 0.05; **P < 0.01; ***P < 0.001 (two-tailed Student's *t*-test). Exact P values are provided in the Source Data file.

differentiate later than interstitial macrophages, in older testes. These results suggest that monocytes that colonize late fetal testes contribute to both adult testicular macrophage populations, with monocytes differentiating into peritubular macrophages in a delayed time window relative to interstitial macrophages.

### Sertoli cells regulate testicular monocyte recruitment and macrophage differentiation

We next assessed which cell types (e.g., Sertoli or germ cells) in the testis microenvironment are important for fetal–monocyte recruitment and differentiation into adult testicular macrophages. We first ablated Sertoli cells by crossing *Rosa*-DTA mice with Sertoli-specific *Amh*-cre mice[43]. In *Amh*-cre; *Rosa*-DTA testes from E14.5 to E18.5, a majority of testis cords with SOX9⁺ or AMH⁺ Sertoli cells were ablated, with only a few testis cords near the gonad–mesonephros border region remaining (Fig. 5a–l), consistent with previous *Amh*-cre; *Rosa*-DTA studies[44,45]. Stereology-based cell counts showed an 80% depletion of Sertoli cells, with no effect on ERG⁺ endothelial cell number; these results were confirmed using qRT-PCR analyses (Supplementary Fig. 11). Cell counts of Leydig cells, whose differentiation is dependent on signals from Sertoli cells, showed that CYP17A1⁺ and HSD3B1⁺ cell number was significantly (35–60%) reduced; these results were consistent with qRT-PCR results (Supplementary Fig. 11).

Given a potential effect on Leydig cells upon Sertoli-cell ablation, we specifically tested a requirement for fetal Leydig cells in immune cell recruitment using *Nr5a1*-cre; *Rosa*-NICD (Notch1

intracellular domain) embryos, which lack differentiated fetal Leydig cells but retain Sertoli cells[46]. In E18.5 *Nr5a1*-cre; *Rosa*-NICD embryos, we observed a significant (~60%) reduction in Leydig cells, both by cell counts and qRT-PCR, but no significant impacts on Sertoli, germ, or endothelial cells (Supplementary Fig. 12a–g, i). We did not observe any significant effects on monocyte/macrophage cell number or gene expression in E18.5 *Nr5a1*-cre; *Rosa*-NICD embryos (Supplementary Fig. 12h, i), indicating that fetal Leydig cells are not a major driver of testicular immune cell recruitment or differentiation.

Although F4/80⁺ macrophage number in Sertoli-cell-depleted testes was comparable to control testes from E14.5 to E18.5 (Fig. 5a–f, m), there was a significant reduction in the number of CD45⁺IBA1⁻ cells (likely mostly monocytes, but could also include other immune cells such as granulocytes) in Sertoli-cell-depleted testes from E16.5 to E18.5 (Fig. 5g–l, n). Cell counts were confirmed using flow cytometry, which showed an overall reduction in testicular CD45⁺ cells caused by fewer F4/80-lo CD11b-hi cells, which represent monocytes or other myeloid cells, in E18.5 *Amh*-cre; *Rosa*-DTA fetal testes versus controls (Fig. 5o–s). These results suggest that Sertoli cells play a major role in fetal testicular monocyte recruitment.

To further examine the role of postnatal Sertoli cells in macrophage differentiation, we analyzed testicular macrophage populations in *Dmrt1*-null knockout (*Dmrt1⁻/⁻*) males[47], which exhibit a postnatal loss of Sertoli-cell identity and the transdifferentiation of SOX9⁺ Sertoli cells to FOXL2⁺ granulosa cells[48,49] (Supplementary Fig. 13a–f). In P7, P18 and P60 *Dmrt1⁻/⁻* testes, CSF1R⁺ interstitial macrophages were still

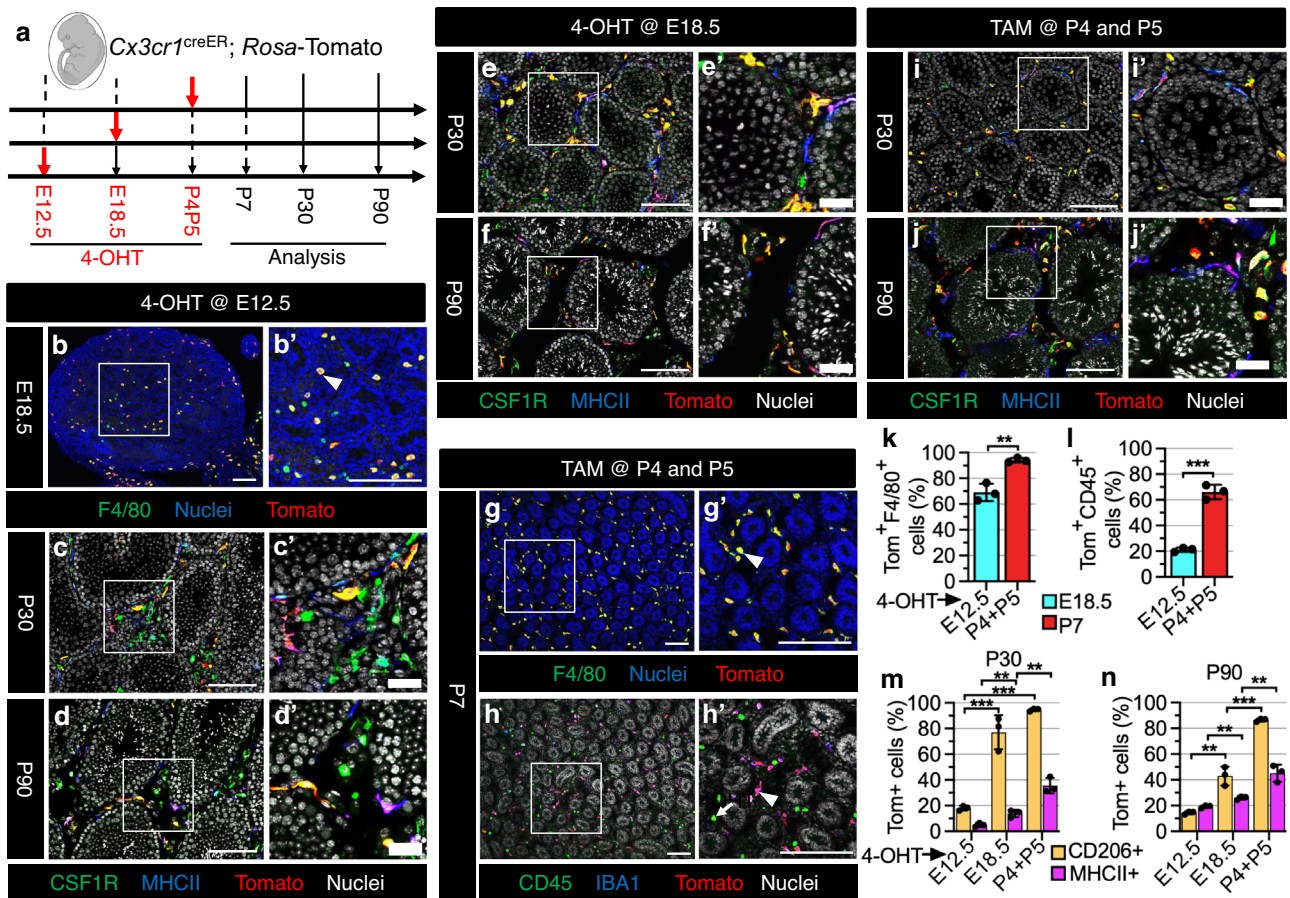

**Fig. 4 | Fetal testis monocytes gradually differentiate into testicular macrophages after birth. a** Strategy for 4-OHT-induced lineage-tracing and harvesting of testes from *Cx3cr1*creER; *Rosa*-Tomato embryos and juvenile/adult mice. The embryo image was created with BioRender.com software (BioRender.com).
**b–j** Representative images (n = 3 independent gonads) of testes at various stages from *Cx3cr1*creER; *Rosa*-Tomato mice exposed to 4-OHT at E12.5 (**b–d**), E18.5 (**e, f**), or to TAM at P4 and P5 (**g–j**). Arrowheads denote Tomato-expressing IBA1[+] or F4/80[+] macrophages and arrows denote Tomato-negative CD45[+] cells. Thin scale bar, 100

μm; thick scale bar, 25 μm. **k–n** Graphs showing quantification (n = 3 independent gonads) of percent Tomato-expressing F4/80[+] macrophages at E18.5 or P7 (**k**), percent Tomato-expressing CD45[+] cells at E18.5 or P7 (**l**), and percent Tomato-expressing interstitial (CD206[+]) and peritubular (MHCII[+]) macrophages at P30 (**m**) or P90 (**n**) in *Cx3cr1*creER; *Rosa*-Tomato testes induced with 4-OHT or TAM at various embryonic or postnatal stages. Data are shown as mean +/− SD. **P < 0.01; ***P < 0.001 (two-tailed Student's t test). Exact P values are provided in the Source Data file.

located in the interstitial compartment similar to *Dmrt1*[+/−] control testes (Supplementary Fig. 13g–l). P7 *Dmrt1*[−/−] testes had normal seminiferous tubules and similar MHCII[+] round cells, similar to P7 control testes (Supplementary Fig. 13g, h); however, at P18, unlike control testes that showed normal MHCII[+] peritubular macrophage localization, which arises postnatally at around 2 weeks of age[23], many MHCII[+] round cells were observed in the interstitial compartment of P18 *Dmrt1*[−/−] testes (Supplementary Fig. 13i, j). By P60, MHCII[+] round cells in mutant testes had developed into macrophages with a stellate or dendritic morphology, but were still retained in the interstitial compartment and were not properly localized to seminiferous tubule surfaces (Supplementary Fig. 13k, l). These results suggest that postnatal Sertoli cells are critical for peritubular macrophage differentiation and localization.

To address whether germ cells have an effect on macrophage recruitment and/or differentiation, we examined fetal and postnatal *Kit*[W/W-v] and *Dnd1*[Ter/Ter] mouse testes, both of which show germ cell deficiency but have otherwise normal seminiferous tubules[50,51]. In E18.5 *Dnd1*[Ter/Ter] fetal testes, we confirmed that normal-shaped testis cords were maintained, but no DDX4[+] (also called MVH) germ cells were detected; in these samples, we observed CD45[+] cells in similar numbers as in control littermates (Supplementary Fig. 14a–d). In P30 *Kit*[W/W-v] and *Dnd1*[Ter/Ter] testes,

we observed normal localization and morphology of CSF1R[+] interstitial and MHCII[+] peritubular macrophages as compared to control testes (Supplementary Fig. 14e–l), suggesting that germ cells do not play a critical role in macrophage recruitment or differentiation.

## EMPs and HSC-derived macrophages have distinct functions in fetal testis development

To address which hematopoietic waves contribute to early fetal testicular macrophages, we exposed *Csf1r*-creER; *Rosa*-Tomato and *Kit*[creER]; *Rosa*-Tomato embryos to 4-OHT at E8.5 and E10.5 to label YS-derived EMPs and AGM-derived HSCs, respectively. In *Csf1r*-creER; *Rosa*-Tomato embryos, there were extensive Tomato-labeled F4/80[+] macrophages (~65–75%) in E12.5 gonad/mesonephros complexes induced at E8.5 or E10.5 (Supplementary Fig. 15a, b, i), whereas CD11b[+] cells (likely monocytes) located in the gonad–mesonephros border region were labeled only after induction at E8.5 (Supplementary Fig. 15c, d). In *Kit*[creER]; *Rosa*-Tomato embryos, a majority of F4/80[+] macrophages and CD11b[+] cells in E12.5 gonad/mesonephros complexes were Tomato-positive when induced at E8.5, but only very rarely (~5%) at E10.5 (Supplementary Fig. 15e–h, j). In addition, some DDX4[+] germ cells were Tomato[+] (Supplementary Fig. 15g, h), consistent with KIT expression in primordial germ cells[52]. These lineage-tracing results suggest that the

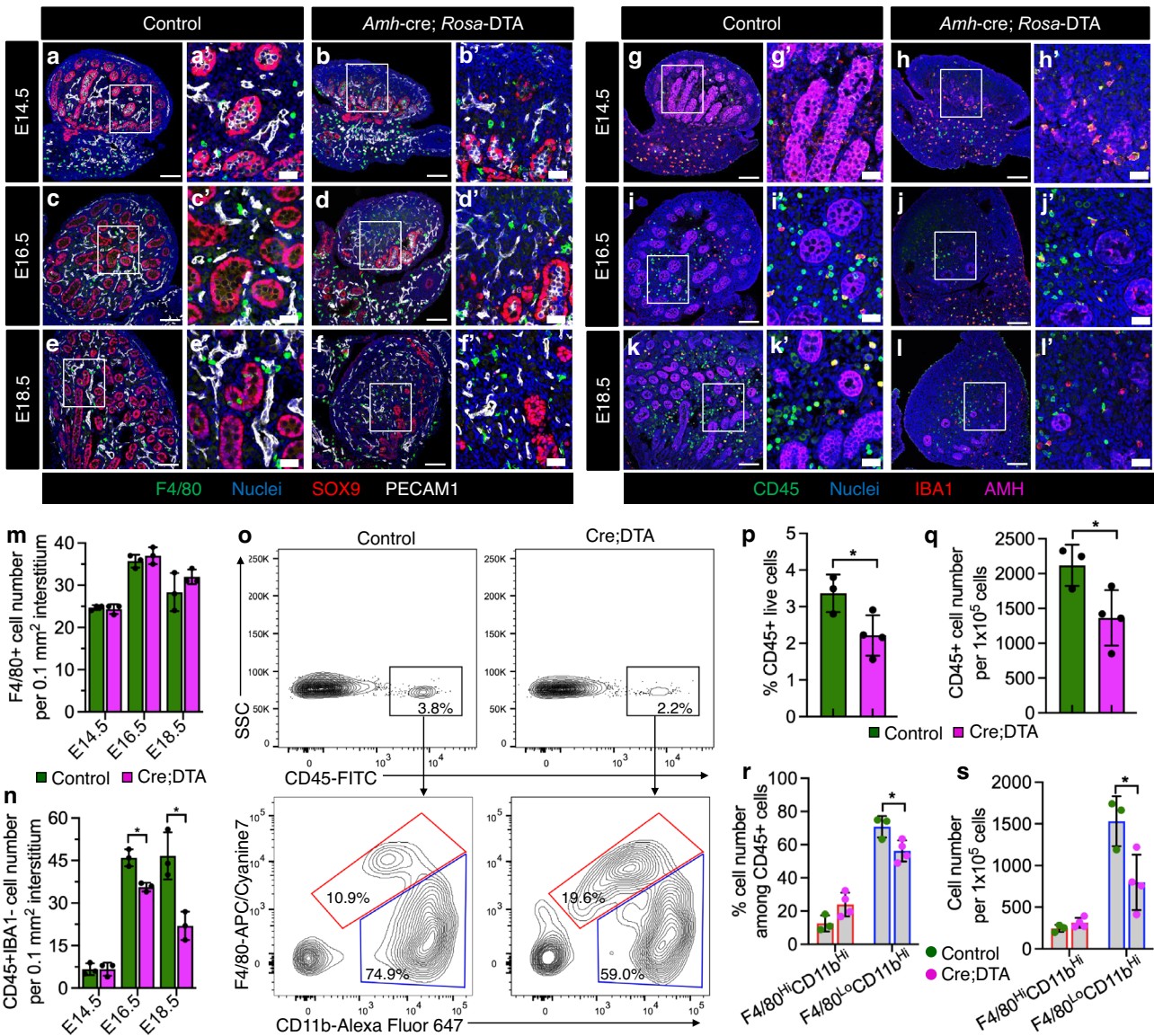

**Fig. 5 | Sertoli cells regulate testicular immune cell recruitment.**
**a**–**l** Representative images ($n = 3$ independent gonads) of control (**a**, **c**, **e**, **g**, **i**, **k**) and
*Amh*-cre; *Rosa*-DTA (**b**, **d**, **f**, **h**, **j**, **l**) testes at various fetal stages. Thin scale bar,
100 μm; thick scale bar, 25 μm. **m**, **n** Graphs showing number ($n = 3$ independent
gonads) of F4/80⁺ cells (**m**) or number of CD45⁺ IBA1-negative cells (**n**) per unit area
of gonadal interstitium at E14.5, E16.5, or E18.5 in control versus *Amh*-cre; *Rosa*-DTA
fetal testes, as determined by stereology-based cell counts. **o** Representative flow
cytometry analyses of E18.5 control (left) and *Amh*-cre; *Rosa*-DTA (right) fetal testes
for CD45⁺ cells (top) and for F4/80-hi CD11b-hi (macrophages; red gate) versus F4/

80-lo CD11b-hi cells (monocytes and other myeloid cells; blue gate) (bottom).
**p**–**s** Graph showing flow-cytometric-based quantification of percent CD45⁺ cells
among live total gonadal cells (**p**), number of CD45⁺ cells per $1 \times 10^5$ live total
gonadal cells (**q**), percent F4/80-hi CD11b-hi and F4/80-lo CD11b-hi cells among
CD45⁺ cells (**r**), and number of F4/80-hi CD11b-hi and F4/80-lo CD11b-hi cells per
$1 \times 10^5$ live total gonadal cells (**s**) for E18.5 control ($n = 3$) versus *Amh*-cre; *Rosa*-DTA
($n = 4$) fetal testes. Data are shown as mean +/− SD. *$P < 0.05$ (two-tailed Student's *t*-
test). Exact *P* values are provided in the Source Data file.

earliest fetal testicular macrophages (i.e., at E12.5) exclusively originate
from YS-derived EMPs.

Since the development and migration of YS-derived macrophage
progenitors are dependent on CSF1R[6], we next specifically depleted
YS-macrophage progenitors by injecting anti-CSF1R antibody at E6.5,
as reported previously[8,53]. CSF1R-antibody injection resulted in effi-
cient depletion of F4/80⁺ macrophages in E13.5 testes (Fig. 6a, b);
however, we observed an increase in the number of CD11b⁺ monocytes
concentrated in the gonad–mesonephros border region (Fig. 6c, d).
qRT-PCR analyses also demonstrated a significant downregulation of
macrophage-related gene expression, such as *Cx3cr1*, *Csf1r*, and *Adgre1*
(also called *F4/80*), in CSF1R-antibody-injected E13.5 gonads compared
to IgG-injected controls (Fig. 6e). qRT-PCR analyses and immuno-
fluorescence assays demonstrated that there was no significant impact

on endothelial cells, Leydig cells, and germ cells in E13.5 gonads
(Fig. 6a–e). Despite no change in the expression of *Sox9* and *Amh* (two
Sertoli-cell genes) in E13.5 CSF1R-antibody-injected gonads (Fig. 6e),
morphometric analyses showed a reduction in testis cord width (but
no effect on testis cord height) (Fig. 6f, g), along with an increase in
abnormal, non-linear testis cords, including fused and branched
cords (Fig. 6h).

We found that testicular macrophages in CSF1R-antibody-
exposed embryos were fully repopulated by E18.5 (Fig. 6i, j), with
extensive CD11b⁺ monocytes present (Fig. 6k, l). In addition,
macrophage-related genes, such as *Cx3cr1*, *Csf1r*, and *Adgre1*, and
monocyte-related genes, such as *Itgam* (encoding CD11b), *Itgb2*
(encoding CD18), *Ccr2*, and *Ptprc* (encoding CD45) were significantly
increased in CSF1R-antibody-injected E18.5 testes compared to

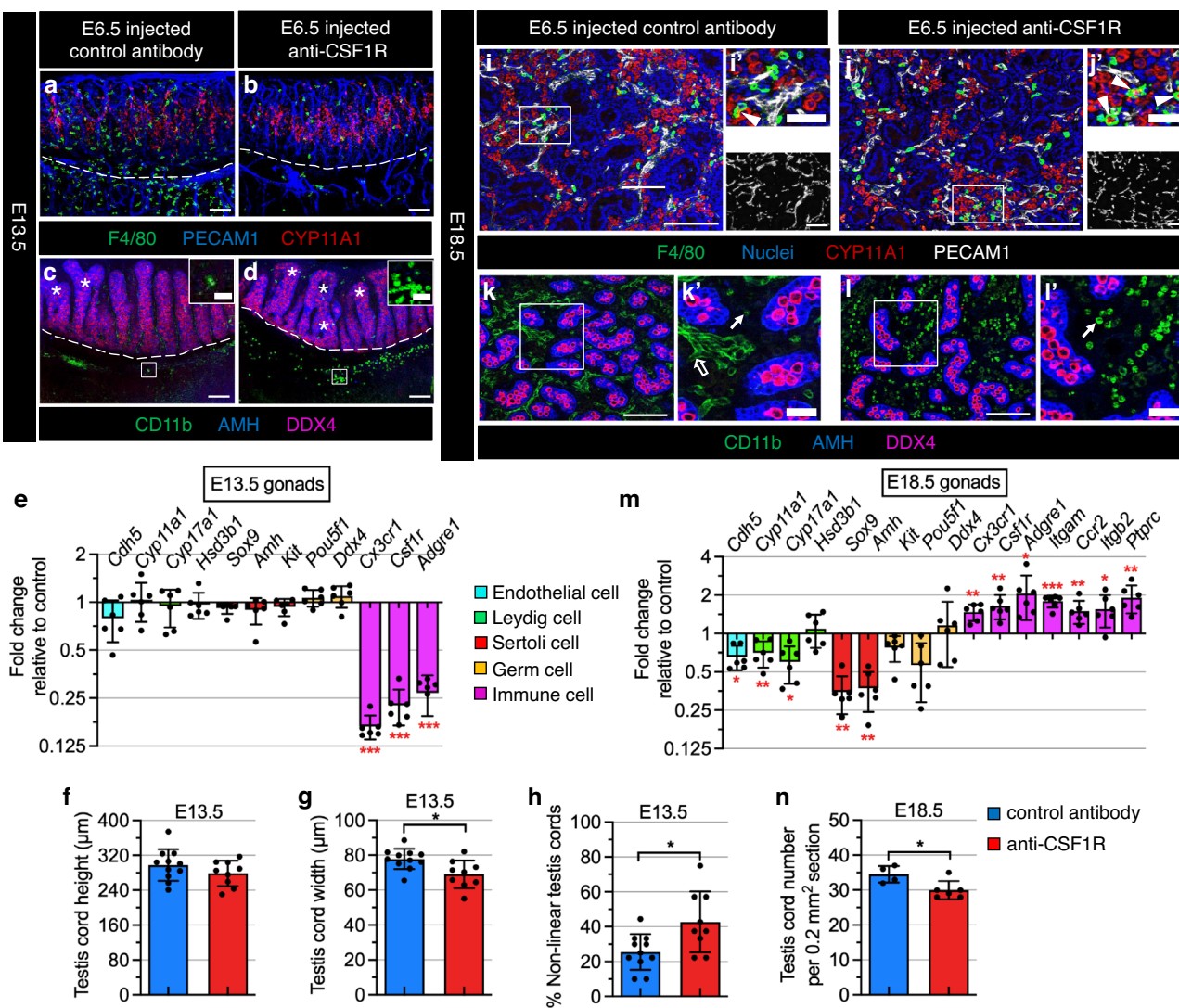

**Fig. 6 | EMPs and HSC-derived macrophages have distinct functions during fetal testis development. a–d** Images of E13.5 fetal testes from C57BL/6 J embryos exposed at E6.5 to either control rat IgG2a antibody (**a**, **c**) or anti-CSF1R blocking antibody to deplete YS-derived macrophages (**b**, **d**). Shown in (**a**, **b**) and (**c**, **d**) are representative images from $n = 4$ and $n = 5$ independent gonads, respectively. Dashed lines indicate gonad–mesonephros boundary. Asterisks denote fused or branched testis cords. Insets in **c** and **d** are higher-magnification images of the boxed regions highlighting CD11b+ cells (likely monocytes) in the gonad–mesonephros border region. Thin scale bar, 100 μm; thick scale bar in **c** and **d** insets, 25 μm. **e** qRT-PCR analyses of whole E13.5 fetal testes showing fold change of gene expression in YS-macrophage-depleted samples ($n = 6$ independent gonads) versus controls ($n = 6$ independent gonads). **f–h** Graphs showing quantification of testis cord height (**f**), testis cord width (**g**), and percent abnormal (fused and/or branched) testis cords (**h**) in E13.5 YS-macrophage-depleted samples ($n = 11$

independent gonads) versus controls ($n = 9$ independent gonads).
**i–l** Representative images ($n = 3$ independent gonads) of E18.5 fetal testes from C57BL/6 J embryos exposed at E6.5 to either control rat IgG2a antibody (**i**, **k**) or anti-CSF1R blocking antibody to deplete YS-derived macrophages (**j**, **l**). Arrowheads denote cells exhibiting co-expression of F4/80 and CYP11A1; white arrows denote CD11b+ monocytes; black arrow denotes the higher background for CD11b in endothelial cells in control samples due to rat IgG2a antibody injection. Thin scale bar, 100 μm; thick scale bar, 25 μm. **m** qRT-PCR analyses of whole E18.5 fetal testes showing fold change of expression in YS-macrophage-depleted samples ($n = 6$ independent gonads) versus controls ($n = 6$ independent gonads). **n** Graph showing E18.5 testis cord number per unit area in YS-macrophage-depleted samples ($n = 6$ independent gonads) versus controls ($n = 4$ independent gonads). All graph data are shown as mean +/– SD. *$P < 0.05$; **$P < 0.01$; ***$P < 0.001$ (two-tailed Student's $t$ test). Exact $P$ values are provided in the Source Data file.

controls (Fig. 6m). These results suggest that YS-derived EMPs are dispensable for giving rise to late fetal testicular macrophages and that alternative hematopoietic precursors could functionally replace them (or at least partially). At E18.5, we observed more CYP17A1+ Leydig cells colocalized with macrophages in CSF1R-antibody-injected testes (Fig. 6i, j), perhaps indicative of phagocytosis, with a reduction in mRNA levels of the Leydig cell genes *Cyp11a1* and *Cyp17a1* (Fig. 6m). E18.5 CSF1R-antibody-treated gonads also had reduced expression of *Cdh5*, *Sox9*, and *Amh* (Fig. 6m), along with more intermittent blood vessels (Fig. 6i, j) and a reduction in the number of AMH+ testis cords (Fig. 6k, l, n). However, mRNA levels of germ cell genes were not

changed (Fig. 6m). These data suggest that repopulated excess testicular macrophages or monocytes after prior CSF1R-antibody-mediated depletion have adverse effects on multiple cell types in late fetal testes.

### Adult interstitial macrophages specifically promote Leydig cell proliferation and steroidogenesis
Given adult interstitial macrophages located within the testicular parenchyma are in close contact with Leydig cells[26,29], we hypothesized that interstitial macrophages specifically regulate adult Leydig cell development and steroidogenesis. To address this hypothesis,

we used anti-CD45 and anti-F4/80 magnetic antibody microbeads to efficiently remove or enrich macrophages from isolated primary testicular interstitial cells (which excluded seminiferous tubules) in vitro via magnetic-activated cell sorting (MACS) from adult *Cx3cr1*[GFP] testes, in which testicular macrophages express GFP[26] (Supplementary Fig. 16a). About 90% of GFP[+] cells expressed CD206 (Supplementary Fig. 16a), demonstrating that macrophages enriched by anti-CD45 or anti-F4/80 microbeads were testicular interstitial macrophages.

We then cultured the pre-separation, CD45[−] (CD45-depleted), CD45[+] (CD45-enriched), and F4/80[+] (F4/80-enriched) interstitial cell populations in vitro for 3 days and 6 days. Flow cytometry results showed that in the CD45[−] fraction almost no CD206[+] testicular macrophages appeared and most cells were CD106[+] (officially named VCAM1, which is expressed in adult Leydig cells[54]), especially on day 6 (Supplementary Fig. 16b, c). Surprisingly, the percentage of CD206[+] testicular macrophages on day 3 in the CD45[+] and F4/80[+] fractions was lower than at the onset of culture, but by day 6 it returned to high levels (Supplementary Fig. 16b, c), indicating a varied proliferative capacity of testicular macrophages and Leydig cells at different time points during co-culture. EdU incorporation analyses indicated a higher percentage of EdU[+]CYP11A1[+] proliferating Leydig cells in all fractions on day 3 than on day 6 (Fig. 7a–i), suggesting that Leydig cell proliferative potential diminished as culture time in vitro was extended. In addition, on day 3 rather than day 6, there was a significant increase in the percentage of EdU[+] Leydig cells in the CD45[+] and F4/80[+] fractions compared to the pre-separation fraction, with an even greater increase when compared to the CD45[−] fraction (Fig. 7a–i), suggesting the initial proliferation of Leydig cells is dependent on testicular macrophages. However, CD206[+] testicular macrophages had an increased proliferative capacity in the CD45[+] and F4/80[+] fractions on day 6 (Fig. 7a–h, j).

To define the role of testicular macrophages in Leydig cell steroidogenesis, we measured testosterone levels in the supernatant of each interstitial cell fraction on day 3 and day 6 of culture. We found testosterone concentrations were significantly reduced in the supernatant of the CD45[−] fraction, and were increased in the supernatant of the CD45[+] and F4/80[+] fractions compared to the pre-separation fraction from day 0 to day 3 (Fig. 7k). However, from day 3 to day 6, Leydig cells produced extremely low levels of testosterone in all fractions, with even lower testosterone levels in the CD45[−], CD45[+], and F4/80[+] fractions, which contained either no macrophages or extensive macrophages (Fig. 7l). These data suggest that testicular macrophages promote Leydig cell steroidogenesis, but that an excess of testicular macrophages may be detrimental to this process during prolonged culture.

To address which testicular macrophage population plays a dominant role in Leydig cell steroidogenesis, we purified interstitial and peritubular macrophages using fluorescence-activated cell sorting (FACS) and co-cultured them with Leydig cells. We found that MHCII[+] peritubular macrophages strongly and exclusively expressed chemokine (C-C motif) receptor 2 (*Ccr2*), as reflected by GFP expression in *Ccr2*[GFP/+] adult testes, whereas CD206[+] interstitial macrophages did not express GFP (Fig. 7m). Thus, we cultured Leydig cells alone (GFP[−]CD206[−]) and Leydig cells in combination with interstitial macrophages (GFP[−]CD206[+]) or peritubular macrophages (GFP[+]CD206[−]) for 3 days in vitro. We detected higher testosterone levels in the supernatant of a co-culture of Leydig cells with interstitial macrophages as compared to a co-culture of Leydig cells with peritubular macrophages, the latter of which was indistinguishable from a Leydig-only culture (Fig. 7n). Our findings suggest that testicular interstitial macrophages, not peritubular macrophages, promote Leydig cell steroidogenesis.

## Discussion

Although it is clear that there are at least two distinct macrophage populations in the adult testis[22,23,26], there has been controversy in recent studies about the hematopoietic origins of testicular macrophages[22–24]. Here, our analyses of multiple lineage-tracing models indicate that in late gestation, monocytes originating from AGM-derived HSCs are recruited into the fetal testis and give rise to the two adult testicular macrophage subsets, CSF1R[+]CD206[+]MHCII[−] interstitial macrophages and CSF1R[−]CD206[−]MHCII[+] peritubular macrophages, albeit with distinct developmental timing. In contrast, YS-derived early EMPs only have a minor contribution to adult macrophages and instead are the source of the initial population of early fetal testicular macrophages that are gradually replaced over the course of life. Unlike other tissues that have been examined, such as the liver[7,8], our results suggest that late EMPs only make a minor contribution to testicular macrophages, and the early- and late-EMP-derived cells of the fetal testis are virtually completely replaced within the organ by early adulthood by fetal–monocyte-derived macrophages. Furthermore, HSCs arising from either the neonatal or adult bone marrow do not contribute to adult testicular macrophages, at least in a normal developmental context.

One caveat to these conclusions is that *Kit*[creER] fate-mapping may not unequivocally trace AGM-derived HSCs but may also label late-EMP-derived HSCs, as proposed by a previous study[11]. However, according to our analyses, it appears that AGM-derived HSCs gradually become the predominant target in the testis when 4-OHT-pulsed at E10.5. Additionally, the migration of YS-derived EMPs towards tissues peaks at around E10.5 and ends around E12.5[32], which is not consistent with our observed onset of testicular monocyte influx starting after E14.5, indicating that YS-derived EMPs are not likely a main source of tissue-resident macrophages in the adult testis. Since there is some debate as to whether or not *Kit*-creER strictly targets AGM-derived HSCs[55], this area of study is still open to further investigation. However, overall our findings support a model in which AGM-derived fetal HSCs, during a limited developmental window, contribute to late fetal and adult testicular macrophage populations, and they address a knowledge gap in our understanding of testis-resident macrophage ontogeny.

Given the unique immunosuppressive environment of the testis, major outstanding questions in the field are how this environment is established and how testicular immune cells are recruited. One significant aspect of this process can be attributed to Sertoli cells, which contribute to the immune privilege of the testis by forming the blood-testis-barrier (BTB)/Sertoli-cell barrier to prevent immune cells from obtaining access to potentially auto-antigenic post-meiotic germ cells[16,56]. Beyond the formation of the BTB, Sertoli cells have immunomodulatory functions and can induce the recruitment of regulatory T cells (Tregs) from naive T cells which, in turn, promotes Sertoli-cell immunosuppressive functions in graft survival[57]. In our study, we demonstrated that Sertoli cells, but not germ cells, are essential for fetal testicular monocyte recruitment and macrophage differentiation. Further investigation is needed into the mechanism of how Sertoli cells recruit immune cells, but our data using the *Amh*-Cre-mediated ablation model indicates that endothelial cells are not impacted and it is unlikely that the decrease of testicular monocyte number in Sertoli-cell-depleted fetal testes is due to impaired testicular vasculature. Instead, we observed a reduction in Leydig cell number, which is consistent with a previous report utilizing fetal Sertoli-cell ablation[58]. Using *Nr5a1*-cre;*Rosa*-NICD embryos, we found that specific reduction of Leydig cells (in the context of normal Sertoli cells) had no significant impact on immune cell numbers. This finding differs from previous studies, in which sustained depletion of adult Leydig cells resulted in long-term reduced numbers of adult testicular macrophages[59]. This difference is likely due to distinct mechanisms of fetal versus adult

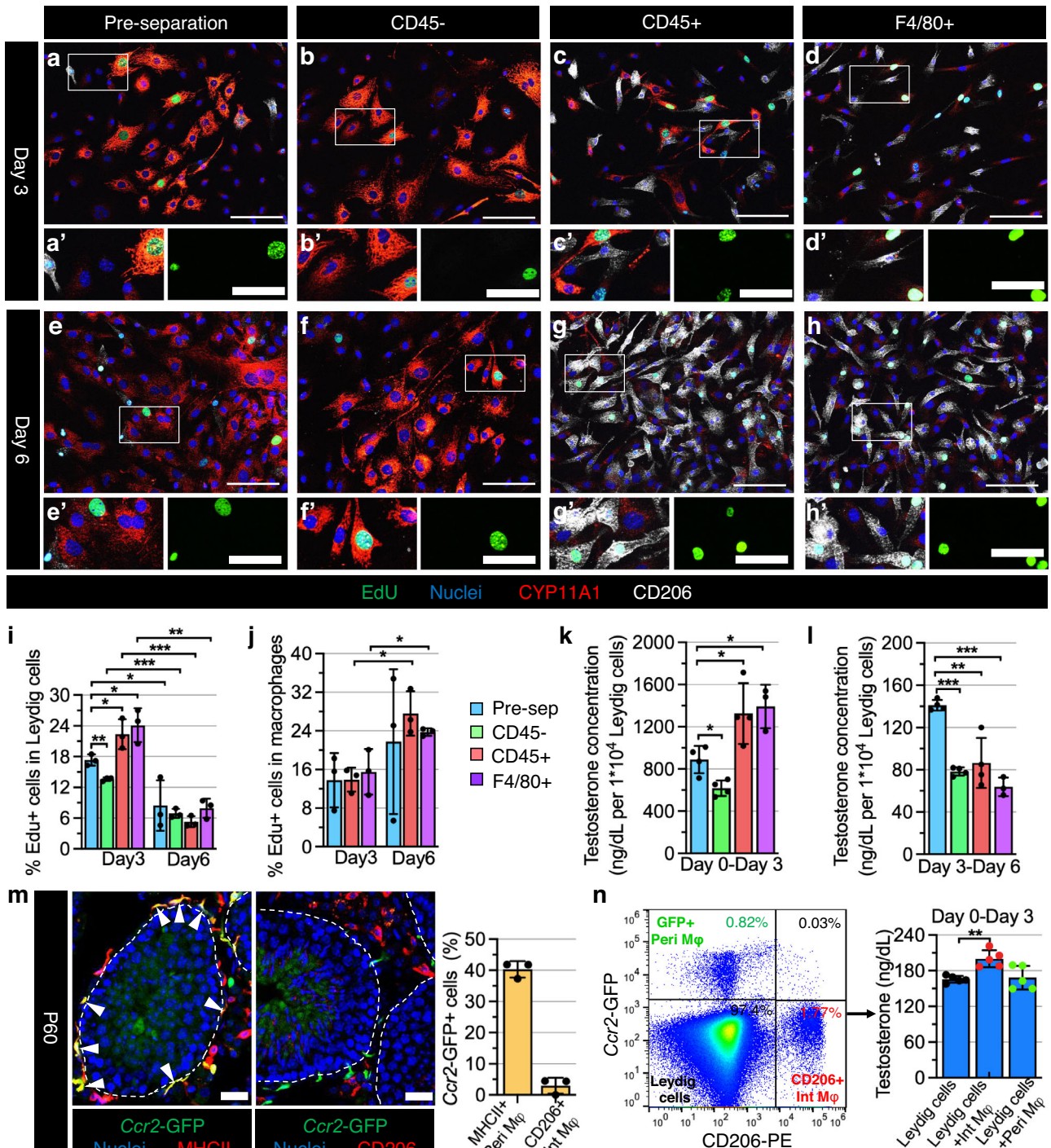

**Fig. 7 | Adult interstitial macrophages promote Leydig cell proliferation and steroidogenesis. a–h** Representative images (*n* = 3 independent experiments) of primary cell culture after 3 days (**a–d**) and 6 days (**e–h**) from adult (3-month-old) C57BL/6 J testes for pre-separation (**a**, **e**), CD45-depleted (**b**, **f**), CD45-enriched (**c**, **g**), and F4/80-enriched (**d**, **h**) populations. **i–l** Graphs showing quantification (*n* = 3 independent experiments from 6 testes) of percent EdU⁺ Leydig cells (**i**), percent EdU⁺ macrophages (**j**), and testosterone concentration (*n* = 4 independent experiments from 8 testes) in culture media after 0–3 days of culture (**k**) and 3–6 days of culture (**l**) in the 4 different cell populations. **m** Representative images of *Ccr2*^GFP/+ testes and graph showing quantification (*n* = 3 independent testes) of percent GFP-expressing interstitial (CD206⁺) and peritubular (MHCII⁺)

macrophages at P60. Arrowheads denote GFP-expressing MHCII⁺ peritubular macrophages. **n** FACS isolation and culture of testicular macrophages from adult *Ccr2*^GFP/+ testes. Flow cytometry plot (left) shows peritubular macrophages (Peri Mφ: GFP⁺CD206⁻), interstitial macrophages (Int Mφ: GFP⁻CD206⁺) and Leydig cells (GFP⁻CD206⁻) gated on total live cells as indicated; graph (right) shows testosterone concentrations (*n* = 5 independent experiments from 10 testes) in culture media after 3 days of culture of Leydig cells alone and Leydig cells co-cultured with Int Mφ or Peri Mφ. Thin scale bar, 100 µm; thick scale bar, 25 µm. All graph data are shown as mean +/– SD. *P < 0.05; **P < 0.01; ***P < 0.001 (two-tailed Student's *t* test). Exact *P* values are provided in the Source Data file.

testicular immune cell development and/or distinct factors secreted by fetal versus adult Leydig cells.

Our results suggested that the loss of Sertoli-cell identity in $Dmrt1^{-/-}$ postnatal testes, rather than germ cell loss, compromised the differentiation of peritubular macrophages, which subsequently had an abnormal interstitial localization. These defects may be due to chemokine and cytokine ligand-receptor systems that are required for macrophage differentiation, migration, and activation[60]. Although monocyte chemoattractant protein-1 (MCP-1; official name CCL2) is detected in Sertoli cells[61] and its receptor, CCR2, is expressed in testicular peritubular macrophages[62], the number and localization of peritubular macrophages are completely normal in $Ccr2^{-/-}$ mice[22,24]. This finding suggests that the MCP-1/CCR2 axis is not required for peritubular macrophage differentiation and distribution in the testis. In addition, the CSF2 (also called GM-CSF)/CSF2R pathway induces M1 polarization of macrophages by increasing MHCII expression and antigen presentation capacity of macrophages[63]. Testicular peritubular macrophages have a high level of MHCII expression and have been classified as M1 macrophages[23,26]. However, whether the CSF2/CSF2R axis plays an important role in the regulation of Sertoli cells during peritubular macrophage development warrants further study.

A previous study using a $Cx3cr1^{cre}$; $Rosa$-DTA-mediated approach showed that early fetal macrophages regulate fetal testis morphogenesis[21]. In this study, we also found that E13.5 testes specifically depleted of YS macrophages (using a CSF1R-antibody-based approach) have an increased number of irregularly branched and fused testis cords. However, we observed numerous CD11b+ myeloid cells in the gonad−mesonephros border region. In a recently published report[25], gonads lacking $Mafb$ and $Maf$ (also called $c$-$Maf$), encoding transcription factors required for monocyte/macrophage development[64–66], also exhibited extensive supernumerary CD11b+ myeloid cells, proposed to be monocytes; however, F4/80+ macrophages were not affected in $Maf$ and $Mafb$ mutant gonads. In addition, testis cords in $Maf$ mutant gonads were smaller, but likely due to a significant decrease in the number of germ cells[25]. Our current results and a previous study[21] showed that macrophage-depleted testes, either by using anti-CSF1R-antibody injection or a $Cx3cr1^{creER}$; $Rosa$-DTA ablation method, have a normal number of germ cells. Thus, a disruption in the ratio between monocytes and macrophages may be a major cause of aberrant testis cords. Although testicular macrophages recovered or even exceeded their normal levels in CSF1R-injected E18.5 testes, as shown in previous reports on other fetal organs including liver, skin, kidney and lung[8], there was still a large number of CD11b+ monocytes with a decrease in the number of testis cords. Therefore, these results indicate that excess CD11b+ monocytes in fetal gonads may have a detrimental effect on testis cord morphogenesis and development, with the ratio of monocytes to macrophages being more important than macrophage quantity.

Macrophages and Leydig cells are in intimate contact throughout testis development and form intercytoplasmic digitations during puberty[67,68], signifying potentially essential roles of testicular macrophages in Leydig cell function. Reports have demonstrated that testicular macrophages promote testosterone production by Leydig cells in vitro;[69] macrophage-deficient $Csf1^{op/op}$ mice are sub-fertile due to low testosterone levels;[70,71] and local depletion of testicular macrophages by intratesticularly injecting dichloromethylene diphosphonate-containing liposomes (Cl2MDP-lp) inhibits Leydig cell proliferation and regeneration[27,72]. However, it is unclear which specific testicular macrophage populations are involved. Our results clearly show that testicular interstitial macrophages, rather than peritubular macrophages, stimulate Leydig cell steroidogenesis in vitro via a co-culture model including varying ratios of Leydig cells and macrophages, which is more similar to the in vivo milieu of tight contact between Leydig cells and macrophages than previous experiments using conditioned culture media[69]. Our previous findings showed that the depletion of

both testicular macrophage populations by a diphtheria-toxin-mediated ablation method reduced intratesticular testosterone levels[26]. In addition, CSF1-Fc-treatment in mice causes an expansion of testicular interstitial macrophage numbers (which specifically express CSF1R) and an increase in circulating testosterone levels[73]. Therefore, testicular interstitial macrophages may be a key, specific immune regulator of Leydig cell steroidogenesis. Adult testicular interstitial and peritubular macrophages are in a non-proliferative state, and they highly express the anti-inflammatory factor $Il10$ and pro-inflammatory factor $Il1b$, respectively[23]. Numerous studies have shown that IL-1B inhibits testosterone synthesis or secretion by Leydig cells in vitro;[74–76] thus, whether non-proliferative interstitial macrophages stimulate testosterone synthesis via IL-10 needs to be investigated in the future. Despite the fact that peritubular macrophages are unrelated or inhibitory to Leydig cell steroidogenesis, they may be a positive regulator of spermatogonial differentiation[26,77]. Therefore, testicular interstitial and peritubular macrophages have unique functions in testis development, even though they are derived from the same fetal hematopoietic progenitors.

Based on our findings regarding testosterone levels and macrophage proliferation in different time windows during in vitro culture, we propose that testosterone may inhibit testicular macrophage proliferation. Indeed, we found that androgen receptor (AR) was expressed in the membrane of testicular macrophages and had a nuclear localization in Leydig cells in vivo and in vitro (Supplementary Fig. 17a, b), indicating that testosterone may inhibit testicular macrophage proliferation through non-genomic pathways[78,79]. Testosterone also inhibits human monocyte/macrophage proliferation and induces the secretion of anti-inflammatory factors in vitro[80,81]. Our results also suggested that a significant increase in testicular macrophages suppressed testosterone production of Leydig cells during extended in vitro culture periods. In $Maf$ mutant fetal gonads, supernumerary monocytes were associated with a down-regulation of many steroidogenic genes[25]. Therefore, a balance between Leydig cell steroidogenesis and testicular macrophage proliferation may be critical for testis development and function.

## Methods

### Mice
All mice used in this study were bred and housed under a 12-h light/12-h dark cycle and specific pathogen-free conditions with ambient temperature (22 °C) and humidity (40–60%) in the Cincinnati Children's Hospital Medical Center's animal care facility, in compliance with institutional and National Institutes of Health guidelines. All experimental procedures were approved by the Institutional Animal Care and Use Committee (IACUC) of Cincinnati Children's Hospital Medical Center (IACUC protocols #IACUC2018-0027 and IACUC2021-0016). $Csf1r$-creER [Tg(Csf1r-Mer-iCre-Mer);1Jwp JAX stock #019098][82], $Cx3cr1^{creER}$ [$Cx3cr1^{tm2.1(creERT2)Jung}$/J; JAX stock #020940][12], $Amh$-cre [Tg(Amh-cre)$^{8815Reb}$/J; JAX stock #033376][43], $Nr5a1$-cre [Tg(Nr5a1-cre)7Lowl/J; JAX stock #012462][83], $Rosa$-Tomato [Gt(ROSA)26Sor$^{tm14(CAG-tdTomato)Hze}$/J; also called Ai14, JAX stock #007914][84], $Rosa$-DTA [Gt(ROSA)26Sor$^{tm1(DTA)Lky}$/J; JAX stock #009669][85], $Cx3cr1^{GFP}$ ($Cx3cr1^{tm1Litt}$/J; JAX stock #005582][86], $Ccr2^{GFP}$ ($Ccr2^{tm1.1Cln}$/J; JAX stock #027619][87], $Kit^{W}/Kit^{W-v}$ compound heterozygous males (WBB6F1/J-Kit$^{W}$/Kit$^{W-v}$/J; JAX stock #100410), and wild-type C57BL/6J (B6) mice (JAX stock #000664) were obtained from The Jackson Laboratory and maintained according to the instructions provided by The Jackson Laboratory. $Kit^{creER}$ [$Kit^{tm2.1(cre/Esr1*)Jmol}$/J] mice[88] (obtained from J. Molkentin, Cincinnati Children's Hospital Medical Center, but are publicly available from Jackson Laboratories; JAX stock #032052), $Flt3$-cre mice[89] [Tg(Flt3-cre)#Ccb; obtained from K. Lavine, Washington University School of Medicine], and $Dmrt1^{-/-}$ mice[47] ($Dmrt1^{tm1.1Zark}$; obtained from D. Zarkower, University of Minnesota) were maintained on a B6 background. $Rosa$-NICD mice[90]

[Gt(ROSA)26Sortm1(Notch1)Dam/J; JAX stock #008159; obtained from J.-S. Park, Cincinnati Children's Hospital Medical Center] were maintained on an ICR background. *Dnd1*[Ter] mice[51] were a gift from Blanche Capel (Duke University Medical Center); *Dnd1*[Ter] mice were originally on a 129T2/SvEmsJ background, but were backcrossed to C57BL/6J for three generations to eliminate the occurrence of testicular teratomas while maintaining complete germ cell depletion[91]. *Dnd1*[Ter/Ter] homozygous mice generated by an intercross of *Dnd1*[Ter/+] heterozygous mice were genotyped by using a Custom TaqMan SNP Genotyping Assay (Thermo Fisher #4332077). Embryonic time points were defined by timed matings, and noon on the day of the appearance of a vaginal plug was considered E0.5.

### Tamoxifen-induced creER activation

To label fetal-derived macrophages, *Cx3cr1*[creER], *Kit*[creER] and *Csf1r*-creER males were crossed with *Rosa*-Tomato reporter females. Pregnant females were intraperitoneally injected at the embryonic stages indicated with 75 μg/g body weight of 4-hydroxytamoxifen (4-OHT) (Sigma-Aldrich #H6278) and supplemented with 37.5 μg/g body weight of progesterone (Sigma-Aldrich #P0130) to prevent fetal abortions due to potential estrogenic effects of tamoxifen. For creER induction in postnatal or adult *Kit*[creER]; *Rosa*-Tomato animals, mice were intraperitoneally injected with 50 μg tamoxifen (TAM) (Sigma-Aldrich #T5648) at P4 and P5; adult mice were treated with 100 μg/g body weight of TAM on three consecutive days from P60-P62.

### Depletion of YS macrophages

To deplete yolk-sac-derived macrophages, pregnant B6 females were treated at E6.5 with a single intraperitoneal injection of 3 mg of anti-CSF1R mAb (Bio X Cell, clone AFS98 #BP0213) or rat IgG2a isotype control (Bio X Cell, clone 2A3 #BP0089) as previously described[8,22].

### Isolation of primary testicular interstitial cells

Interstitial cells were isolated from adult (9–12 weeks old) wild-type B6, heterozygous *Cx3cr1*[GFP] or heterozygous *Ccr2*[GFP] mice following a previously described protocol[92]. Briefly, decapsulated testes were digested with RPMI-1640 medium (Sigma #R5158) containing 0.25 mg/mL collagenase IV (Worthington #LS004186), 100 μg/ml DNase I (Sigma #10104159001) and 2% heat-inactivated fetal bovine serum (FBS; Thermo Fisher #16000044) in a 120-rpm shaker at 34 °C for 10 min. FBS was added until the final concentration reached 10% to stop the enzymatic reaction. A single-cell suspension was obtained by removing seminiferous tubules with a 100-μm cell strainer (Corning #352360). The supernatant was pelleted, washed with RPMI-1640 medium, and incubated in ACK (Ammonium-Chloride-Potassium) buffer (Life Technologies #A10492-01) for 3 min at room temperature to lyse erythrocytes. The cell suspension was pelleted, washed with RPMI-1640 medium and kept in RPMI-1640 medium containing 2% FBS for subsequent analyses.

### In vitro culture of testicular macrophage-enriched interstitial cells by MACS sorting

A dissociated single-cell suspension was passed through a 30-μm nylon mesh (Pre-Separation Filters; Miltenyi Biotec #130-041-407) to remove cell clumps that could clog the column. After counting cell numbers, an appropriate number of cells was pipetted, which was defined as pre-separation interstitial cells. The remaining cells were pelleted and resuspended in 80 μl of MACS buffer (PBS pH 7.2, 0.5% BSA and 2 mM EDTA) and 20 μl anti-CD45 MicroBeads (Miltenyi Biotec #130-052-301) or 20 μl anti-F4/80 MicroBeads (Miltenyi Biotec #130-110-443) per 10^7 total cells. The mixture was incubated in the dark at 4 °C for 15 min, washed, and resuspended in MACS buffer. Cell suspension was applied onto the pre-balanced LS column (Miltenyi Biotec #130–097-679) that was placed in the magnetic Separator. Then the LS column was removed from the separator and the magnetically labeled cells were

immediately flushed out, which were defined as the CD45[+] or F4/80[+] fraction. After incubation with anti-CD45 microbeads, unlabeled cells which passed through the LS column and LD column (Miltenyi Biotec #130-042-901) were defined as the CD45[-] fraction. Each fraction was seeded in 24-well plates with or without glass coverslips and maintained in a humidified atmosphere (5% $CO_2$, 95% air) at 37 °C for 3 days or 6 days in culture media (10% FBS in RPMI-1640).

### Flow cytometry

E18.5 fetuses were rapidly harvested and washed in PBS to remove maternal or placental blood prior to euthanasia. To collect fetal blood, fetuses were decapitated in a 60-mm Petri dish filled with HBSS buffer (Sigma #H6648) containing 25 mM HEPES (Sigma #H0887) and 30 IU/ml heparin (Sigma #H4784). After being allowed to bleed for 5 min, collected blood was incubated in ACK lysis buffer to remove red blood cells. E18.5 livers and testes were enzymatically digested with RPMI-1640 medium containing 1 mg/mL collagenase IV, 100 μg/ml DNase I, and 2% FBS at 37 °C for 20 min to obtain single-cell suspensions. Each interstitial cell fraction in in vitro culture on day 3 or day 6 was digested with Accumax (STEMCELL Technologies #07921) for 5 min at 37 °C. Total cell number was determined in order to calculate the actual number of Leydig cells and macrophages based on percentages provided by the below flow cytometry analysis. Cell suspension was pelleted and washed with FACS buffer (2 mM EDTA, 2% FBS in PBS). Cells were incubated with an Fc blocker antibody (anti-CD32/16; BioLegend #101320) for 10 min and then incubated with the respective mix of antibodies (Supplementary Table 1) in FACS buffer for 30 min at 4 °C. For viability staining, the labeled cells were washed and incubated with Zombie UV Fixable Viability Kit (BioLegend #423107) or Hoechst 33342 (1 mg/ml; Thermo Fisher #H1399) in FACS buffer before analysis. Flow cytometry analysis was performed on a BD Biosciences Fortessa flow cytometer. Viability staining and singlet profiling were always used to pre-gate cells. Basic gating strategies used for flow cytometry experiments are provided in Supplementary Fig. 18.

### In vitro co-culture of interstitial or peritubular macrophages with Leydig cells via FACS

Interstitial cells from heterozygous *Ccr2*[GFP] mice were isolated according to the above flow cytometry method for CD206 and Zombie UV staining. Then cells were pelleted and resuspended in 500 μl sorting buffer (2 mM EDTA, 2% FBS, 25 mM HEPES in HBSS). After gating singlet and live cells, CD206[+]GFP[-] interstitial macrophages, CD206[-]GFP[+] peritubular macrophages and CD206[-]GFP[-] Leydig cells were sorted into 1.5-ml Eppendorf tubes using a MA900 cell sorter (Sony). To mimic the proportion of testicular macrophages observed in pre-sorted interstitial cells, $2 \times 10^5$ Leydig cells alone and $2 \times 10^5$ Leydig cells in combination with 5,000 interstitial macrophages (2.5%) or 5,000 peritubular macrophages (2.5%) each well were plated in 48-well collagenase-coated plates (Corning #354505) in RPMI-1640 medium containing 10% FBS for 3 days in vitro.

### Immunofluorescence

E12.5 and E13.5 gonads were dissected in PBS and fixed overnight in 4% paraformaldehyde (PFA) with 0.1% Triton X-100 at 4 °C for whole-mount immunofluorescence as previously described[21]. Testes of E14.5, E16.5, E18.5, P7, P18, P30, P60, P90, and P120 mice were dissected in PBS and fixed overnight in 4% PFA with 0.1% Triton X-100 at 4 °C; after overnight fixation, testes were processed through a sucrose:PBS gradient (10%, 15%, 20% sucrose) and were embedded in OCT medium (Fisher Healthcare, #4585) at −80 °C prior to cryosectioning, as previously reported[50]. After cryosectioning, samples were washed several times in PBTx (PBS + 0.1% Triton X-100) and incubated in a blocking solution (PBTx + 10% FBS + 3% bovine serum albumin [BSA]) for 1–2 h at

room temperature. Primary antibodies used for immuno-fluorescence were diluted in blocking solution according to Supplementary Table 1 and applied to samples overnight at 4 °C. After several washes in PBTx, fluorescent secondary antibodies conjugated with Alexa Fluor 488, Alexa Fluor 555, Alexa Fluor 647, or Cy3 (Molecular Probes/Thermo Fisher or Jackson Immunoresearch, all at 1:500 dilution) and nuclear dye (2 mg/ml Hoechst 33342) were diluted in blocking solution and applied to cryosections or interstitial cells for 1 h and to whole gonads for 2–3 h at room temperature. Samples were imaged either on a Nikon Eclipse TE2000 microscope (Nikon, Tokyo, Japan) with an Opti-Grid structured illumination imaging system using Volocity software (PerkinElmer, Waltham, MA, USA) or on a Nikon A1 Inverted Confocal Microscope (Nikon, Tokyo, Japan).

### EdU incorporation assays
Interstitial cell proliferation in in vitro culture was determined by the Click-it EdU Alexa Fluor 488 Kit (Invitrogen/Thermo Fisher #C10337) according to the manufacturer's instructions. For co-staining of EdU, CYP11A1 and CD206, interstitial cells plated on glass coverslips were incubated in culture media containing 10 μM EdU for 2 h at 37 °C and then fixed in 4% PFA for 10 min on ice followed by permeabilization with 0.5% Trion X-100 for 20 min at room temperature. After several washes in PBS containing 3% BSA, interstitial cells were incubated with Click-iT reaction cocktail for 30 min at room temperature. CYP11A1 and CD206 primary antibodies, as well as the corresponding secondary antibodies, were then incubated according to the above cryosectioning protocol.

### Quantitative real-time PCR (qRT-PCR)
RNA extraction, cDNA synthesis, and qRT-PCR were performed as previously described[50]. Total RNA was obtained with TRIzol Reagent (Invitrogen/Thermo Fisher #15596018). Genomic DNA was digested with DNase I (Amplification Grade; Thermo Fisher #18068015) to purify RNA. An iScript cDNA synthesis kit (BioRad #1708841) was used on 500 ng of RNA for cDNA synthesis, as per the manufacturer's instructions. qRT-PCR was performed using the Fast SYBR Green Master Mix (Applied Biosystems/Thermo Fisher #4385616) on the StepOnePlus Real-Time PCR system (Applied Biosystems/Thermo Fisher; software version 2.3). Expression levels were normalized to *Gapdh*. Relative fold changes in gene expression were calculated relative to controls using the $2^{-\triangle\triangle Ct}$ method. Primers used for qRT-PCR analysis are listed in Supplementary Table 2.

### Hormone measurements
Testosterone measurements of cell culture media were performed by the University of Virginia Center for Research in Reproduction Ligand Assay and Analysis Core. Interstitial cell supernatants were collected on day 3 and day 6 of in vitro culture, centrifuged at $2000 \times g$ for 5 min at room temperature, and stored at −80 °C until analyzed.

### Cell counts and testis cord morphometric analyses
For all quantifications, images within a field of view $672 \times 900\,\mu m$ (Nikon Eclipse TE2000 microscope) or $640 \times 640\,\mu m$ (Nikon A1 Inverted Confocal Microscope) for each genotype or treatment were analyzed using ImageJ software (NIH). Testis cords of E13.5 and E18.5 XY gonads were visualized by staining with anti-AMH antibody. For E13.5 XY gonads, five testis cords of each whole-mount gonad ($n = 9$ gonads from independent embryos from three independent control-antibody-injected litters; $n = 11$ gonads from independent embryos from four independent anti-CSF1R-antibody-injected litters) were measured and averaged for height and width measurements. All testis cords of each E13.5 whole-mount gonad were counted and used for calculating the percentage of irregular testis cords. For E18.5 XY gonads, all testis cords in each image were counted from at least three

different cryosections (transverse cross-sections) per testis, from $n = 4$ for control-antibody-injected or $n = 6$ for anti-CSF1R-antibody-injected testes. For fetal or adult testicular macrophage/immune cell and Tomato$^+$ cell counting, the Cell Counter plug-in in ImageJ was used to manually count positive cells each image; quantifications were taken from at least three different cryosections per testis, from $n \geq 3$ testes, each from an independent biological replicate. E14.5, E16.5, and E18.5 testis areas were outlined using the Polygon Selection tool, and the area in pixels was calculated with the Measure function; pixel area was then converted to square millimeters based on the pixel dimensions of each image.

### Statistics and reproducibility
Statistical details of experiments, such as the exact value of $n$, what $n$ represents, precision measures (mean ± SD), and statistical significance can be found in Figure Legends. All data are from two to three independent experiments, except for in vitro cell culture from four to five independent experiments. For qRT-PCR, statistical analyses were performed using Prism version 8.0 (GraphPad) and an unpaired, two-tailed Student $t$ test was performed to calculate $P$ values based on ΔCt values. At least two gonads from independent embryos were pooled for each biological replicate ($n \geq 3$ biological replicates) in qRT-PCR analyses. For immunofluorescence, at least three sections from at least $n = 3$ individual animals (i.e., $n = 3$ independent biological replicates) were examined for each time point (i.e., fetal stage or age) and/or experimental condition. For cell culture, at least $n = 4$ independent biological replicates from individual animals were examined for each time point, and each cell population was cultured in 2–3 wells of 24- or 48-well plates for each biological replicate. For flow cytometry analyses, data from 2–3 independent experiments were analyzed using FACS Diva software (BD Biosciences) and FlowJo (BD Biosciences). For cell counting and morphometric analyses, sample sizes are listed above for each group. Graph results are shown as value ± SD, and statistical analyses were performed using an unpaired, two-tailed Student's $t$-test.

### Reporting summary
Further information on research design is available in the Nature Portfolio Reporting Summary linked to this article.

## Data availability
Source data for quantitative assays (e.g., graphs) in Figs. 1–7 and Supplementary Figs. 1–3, 5, 6, 8–12, and 15 are provided with the paper in the Source Data file. Other data supporting the findings of this study are available within the paper and its supplementary information files, and they are also available from the corresponding author upon request. Source data are provided with this paper.

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

## Acknowledgements

We thank Drs. Jeffery Molkentin, Kory Lavine, David Zarkower, Joo-Seop Park, and Blanche Capel for mice and Dr. Dagmar Wilhelm for anti-CYP11A1 antibody. This work was supported by Cincinnati Children's Research Innovation and Pilot Funding (T.D.) and by the National Institutes of Health (grants R35GM119458 and R01HD094698 to T.D.). We also acknowledge BioRender (BioRender.com) for the use of images.

## Author contributions

X.G. conducted experiments, performed data analyses, and co-wrote and edited the manuscript. A.H. conducted experiments and edited the manuscript. S.-Y.L. conducted experiments and edited the manuscript. T.D. supervised the project and co-wrote and edited the manuscript.

## Competing interests

The authors declare no competing interests.
