## [Peer Review File · Nature Communications]

Testicular macrophages are recruited during a narrow fetal time window and promote organ-specific developmental functionsEditorial Note: Figures have been redacted from this file at the request of the authors.

REVIEWER COMMENTS

Reviewer #1 (Remarks to the Author):

In the manuscript by Gu et. al, the authors address a lingering question regarding tissue resident macrophage development and the role of testicular macrophages in promoting organ specific functions. They perform extensive fate-mapping experiments using multiple models to dissect the recruitment windows for yolk sac, fetal monocyte, and bone marrow monocyte progenitor recruitment into distinct testicular macrophage lineages. The authors extend their prior observation that yolk-sac progenitor contributed to testicular macrophage lineages, to now include a primary contribution of fetal-monocytes. They find Sertoli cells supported the differentiation of peritubular macrophages. Further, they observe unique functional characteristics by macrophage subsets in supporting testosterone production in vitro. Together these data significantly advance knowledge of testicular macrophage ontogeny and function. The study is very beautifully performed and data appears to be high quality. However, I do have some remaining questions, particularly regarding some controls and additional data that might help a reader interpret results.

It would be helpful to show for that blood monocytes are not being labeled when sacrificed outside of the immediate TAM treatment window for fate mapping experiments. Since they are short lived and express Csf1r/CX3CR1, they may be a good surrogate for unwanted off-target labeling in stem cells. Based on the data provided, it seems like the Csf1r-creERT2 model will show considerable leakage, whereas the CX3CR1-creERT2 may be restricted.

There has been considerable debate regarding the use of TAM and fate-mapping model influence on normal tissue physiology. These come from studies reporting (i) reporter alleles becoming active independent of tamoxifen treatment (DOI: 10.1038/s41590-018-0272-2), (ii) some cells are prone for death if tamoxifen is provided during high proliferation phase (<https://doi.org/10.1016/j.immuni.2019.09.010>), and (iii) a pre-print suggesting that tamoxifen may enhance embryonic macrophage numbers when treated in utero (<https://doi.org/10.1101/296749>). Providing data to address some of these possibilities would likely strengthen the data. At minimum, it might be helpful to know if the authors observe changes in testicular macrophage numbers in animals given tamoxifen embryonically. These data would likely also be valuable to the field that is currently debating this topic.

While the imaging data is beautiful and helps with localization of cell populations, inclusion of additional quantitative analysis approaches would help to strengthen the story because occasionally it is difficult to determine a conclusion from a narrow field of view. Image analysis or flow cytometric data would be welcomed. In addition, all bar plots should changed to show replicates using symbols.

Fig 5 A-L, can the ablation percentage of Sertoli cells and blood vessels be quantified? Does this result in changes in testosterone levels? Do Sertoli cells express chemokines or maintenance factors to regulate monocyte recruitment (such as CCL2) or macrophage survival (such as Csf1) in the tissue?

The authors need to provide specifics regarding the number of experimental replicates performed for each assay and number of fully independent experiments. Representative plots are shown for a large number of panels, so it would be important to know the replication info, perhaps in the figure legends. I realize some of this information was present in the methods section, but it was unclear if this addressed for all of the experiments in the paper.

Reviewer #2 (Remarks to the Author):

This paper demonstrates that adult testicular macrophages are derived from fetal hematopoietic stem cells (HSCs) using multiple lineage tracing experiments. Authors further claim that the Sertoli cells are essential for the 1) recruitment of monocytes in fetal testis and 2) differentiation of peritubular macrophages in postnatal testis. In addition, testicular macrophages' role in testis' prenatal development has been shown. The article is well written, and the diverse experiments support most conclusions. Although this is an elegant work, mouse adult testicular macrophages' ontogeny was largely deciphered in two recent studies (<https://doi.org/10.1073/pnas.2013686117>; <https://doi.org/10.1038/s41467-020-18206-0>). Like this study, both previous studies also concluded that adult testicular macrophages are derived mainly from fetal liver monocytes with minimal contribution from yolk sac progenitors and adult HSCs, which takes away some of the novelty of this study.

However, the study remains exciting because the authors attempted to demonstrate the involvement of Sertoli cells in the recruitment of monocytes in fetal testis and the differentiation of peritubular macrophages in postnatal testis.

Specific comments that need to be addressed:

Major concern

1) In figure 2, the authors used the KitcreER fate mapping mouse model to demonstrate that adult testicular macrophages primarily originate from AGM-derived HSCs. Their observation is mainly based on Sheng et al., 2015. (<https://doi.org/10.1016/j.immuni.2015.07.016>) work. However, given late EMPs and AGM temporal promiscuity, KitcreER is not an ideal model to distinguish between late EMPs and AGM-derived HSCs. When pulsed with tamoxifen at E8.5, equal labeling of E10.5 EMPs and AGM multipotent progenitors (MPPs) has been reported (Sheng et al., 2015; C Stremmel et al., doi: 10.1038/s41467-017-02492-2). A similar observation was observed in this paper in Figure (2B, 20, 2N), and the authors concluded that "suggesting induction at E8.5 targets YS--derived EMPs and AGM derived HSCs".

It is well established that late EMP-derived fetal liver monocytes constitute the main precursor of adult macrophage populations, including Liver Kupffer cells. However, in supplementary Figure 4D, pulse labeling at E10.5, most of the F4/80+ Kupffer cells are robustly labeled with tomato (not mentioned in the main text). For the reasons stated above, I believe that the KitcreER fate-mapping model will not trace exclusively AGM-derived HSCs, but it may trace both late EMPs as well as AGM-derived HSCs (DOI:<https://doi.org/10.1016/j.immuni.2015.11.022>, doi: 10.1016/j.immuni.2016.02.024). Thereby, I believe that the data presented in this study do not support the claim that adult testicular macrophages are largely derived from AGM- HSCs.

2) To demonstrate the role of Sertoli cells in the recruitment of fetal testicular monocytes, the authors depleted Sertoli cells by using the AMHCre-RosaDTX model. They denoted F4/80+ cells as macrophages and CD45+lba1- as monocytes, which is factually incorrect. lba1 is a marker of macrophages, and lba1- negative cells should not be termed monocytes. It may be a combination of other immune cells, monocytes, neutrophils, and T cells.

From E14.5 until birth, testis contains two macrophage populations: F80hi and F4/80int, with varying proportions (<https://www.nature.com/articles/s41467-020-18206-0>). F4/80Hi cells correspond to the yolk-sac-derived, and F4/80Int cells to the fetal liver monocyte-derived macrophages (doi: 10.1016/j.immuni.2015.03.011). If Sertoli cells recruit fetal monocyte to the testis, the proportion of F4/80Int cells should reduce in the AMHCre-RosaDTX mice compared to the control.

Immunofluorescence analyses will not distinguish between F80hi and F4/80int cells. To examine conclusively whether Sertoli cells recruit fetal monocytes to the testis, authors should perform flow cytometry analysis using macrophage and monocyte-specific markers. This experiment should also be performed in the testis of Dmrt1—mouse.

3) Ablation of the Sertoli cell population significantly reduced Leydig cell number (DOI: 10.1210/en.2017-00196). Because Leydig cells are necessary for the maintenance of testicular macrophages, aberrant change in the number of macrophages/monocytes in Sertoli cell depleted the Leydig cell number may indirectly influence testis. To rule out this possibility, the Leydig cell number should be determined by Stereology in the AMHCre-RosaDTX and Dmrt-/- -mice.

Minor concern

4) Page 3, line 55-58; «... Furthermore, it has also been shown that a number of organs, such as the brain, maintain their tissue-resident macrophage populations (microglia) independently of bone-marrow-derived HSCs, while other organs, such as the gut, exhibit a heavy reliance on bone-marrow HSC-derived monocytes to maintain homeostasis of their tissue-resident macrophages». This sentence does not include information about fetal liver monocytes (FL-Mos) derived macrophages. Thus, the sentence should be modified accordingly.

5) Page 4, lines 63-68; «...The first wave, called primitive hematopoiesis, takes place in the embryonic yolk sac (YS) blood islands starting at embryonic day (E) day (E) 7.0 in mice and gives rise to primitive YS--derived macrophages. Between E8.0 and E8.5, the second wave of hematopoiesis from YS hemogenic endothelium, sometimes referred to as the transient definitive wave, sometimes referred to as the transient definitive wave, gives rise to a class of progenitors called erythromyeloid precursors (EMPs) »

After blood circulation starts at E8.5, EMPs migrate to the fetal liver and expand in number to differentiate into monocytes and other hematopoietic lineages». The first wave of hematopoiesis at E7-7.5 give rise to early erythromyeloid precursors (EMPs) and the second wave of hematopoiesis at E8-8.5 give rise to late erythromyeloid precursors (EMPs). Authors should modify the above sentences for better clarity for the readers.

6) On Page 4, lines 64-75, important references are missing.

Reviewer #3 (Remarks to the Author):

The study evaluates the origin of testicular macrophages using elegant cell tracking methods. This is a very elegant and well designed study that provides new information on the development of tissue macrophages, it challenge old concepts and provides new information.

I suggest its publication

Minor comments:

the discussion is too long and could be edited.

Dear Reviewers,

We are very grateful for your positive response and enthusiasm for our manuscript entitled “**Testicular macrophages are recruited during a narrow fetal time window and promote organ-specific developmental functions**”. We made a concerted effort to address your critiques in the resubmission of our manuscript: we performed additional quantitative assays, including flow cytometry and stereology-based cell counting for multiple experiments; included additional control experiments to assess baseline reporter activity and the effect of tamoxifen on fetal macrophage numbers; specifically addressed the role of fetal Leydig cells in immune cell recruitment; and modified the text to provide clarifications and shorten the Discussion. We found your suggestions to be extremely helpful and, by making these requested changes, we feel that our revised manuscript is much stronger. Reviewer comments are in italics and our responses are listed below in normal blue font.

Reviewer #1:

In the manuscript by Gu et. al, the authors address a lingering question regarding tissue resident macrophage development and the role of testicular macrophages in promoting organ specific functions. They perform extensive fate-mapping experiments using multiple models to dissect the recruitment windows for yolk sac, fetal monocyte, and bone marrow monocyte progenitor recruitment into distinct testicular macrophage lineages. The authors extend their prior observation that yolk-sac progenitor contributed to testicular macrophage lineages, to now include a primary contribution of fetal-monocytes. They find Sertoli cells supported the differentiation of peritubular macrophages. Further, they observe unique functional characteristics by macrophage subsets in supporting testosterone production in vitro. Together these data significantly advance knowledge of testicular macrophage ontogeny and function. The study is very beautifully performed and data appears to be high quality. However, I do have some remaining questions, particularly regarding some controls and additional data that might help a reader interpret results.

We thank the reviewer for their positive comments about our manuscript. In our revised manuscript, we have included the requested controls and additional data to support our claims, as is outlined in our responses below.

It would be helpful to show for that blood monocytes are not being labeled when sacrificed outside of the immediate TAM treatment window for fate mapping experiments. Since they are short lived and express Csf1r/CX3CR1, they may be a good surrogate for unwanted off-target labeling in stem cells. Based on the

*data provided, it seems like the *Csf1r-creERT2* model will show considerable leakage, whereas the *CX3CR1-creERT2* may be restricted.*

We agree that it would be helpful to assess in greater detail which cells are labeled during fate mapping experiments. We used flow cytometry to assess Tomato expression in Ly6C⁺ monocytes and Ly6G⁺ neutrophils, which are typically short-lived cell types, as well as KIT⁺ HSCs, in blood and fetal liver from E18.5 *Csf1r-creER;Rosa-Tomato* and E18.5 *Cx3cr1-creER;Rosa-Tomato* embryos exposed to 4-hydroxytamoxifen (4-OHT) at E12.5. Consistent with the reviewer's prediction, we found that there was some Tomato labeling of monocytes and neutrophils by *Csf1r-creER*, likely due to targeting of KIT⁺ cells (HSCs), but there was virtually no Tomato labeling (1% or less) of monocytes and neutrophils in the *Cx3cr1-creER* model. We have included these new results in Supplementary Figures 5 and 10.

There has been considerable debate regarding the use of TAM and fate-mapping model influence on normal tissue physiology. These come from studies reporting (i) reporter alleles becoming active independent of tamoxifen treatment (DOI: 10.1038/s41590-018-0272-2), (ii) some cells are prone for death if tamoxifen is provided during high proliferation phase (<https://doi.org/10.1016/j.immuni.2019.09.010>), and (iii) a pre-print suggesting that tamoxifen may enhance embryonic macrophage numbers when treated in utero (<https://doi.org/10.1101/296749>). Providing data to address some of these possibilities would likely strengthen the data. At minimum, it might be helpful to know if the authors observe changes in testicular macrophage numbers in animals given tamoxifen embryonically. These data would likely also be valuable to the field that is currently debating this topic.

We concur that these points are important to address. We performed the following:

(i) We assessed the baseline activity of the *Rosa-Tomato* reporter allele in the absence of 4-OHT treatment via immunofluorescence of fetal testes from vehicle-only-injected (ethanol+oil) E18.5 *Csf1r-creER;Rosa-Tomato* embryos pulsed at E12.5, as compared to 4-OHT injection; furthermore, we also examined testes from creER-negative;*Rosa-Tomato* embryos with 4-OHT injection. We failed to detect any Tomato expression in control (i.e., either vehicle-only-treated or creER-negative) testes, indicating there was no spurious activation of reporter alleles. These new data are included in Supplementary Figure 2.

(ii) To address the possibility of increased cell death or aberrant cell cycle activity after 4-OHT administration, we examined cleaved Caspase 3 and MKI67 expression via immunofluorescence of fetal testes from vehicle-only-injected versus 4-OHT-injected E13.5 *Csf1r-creER;Rosa-Tomato* embryos pulsed at E12.5; furthermore, we also examined testes from creER-negative;*Rosa-Tomato* embryos with 4-OHT injection. We observed no significant changes in MKI67 expression patterns in macrophages or any other cell types caused by creER activity or tamoxifen administration, and quantitative stereology-based cell counting of cleaved-Caspase-3-positive cells revealed no differences in apoptotic cell number between any of the genotypes or treatment conditions. In conjunction with other new data showing similar counts of CD45⁺ cells in these testes (see next paragraph), our data suggest that there is no rapid, acute effect of 4-OHT administration or creER expression/activity on cell cycle or apoptosis in fetal testicular macrophages. These new data are included in Supplementary Figure 1.

(iii) To address the hypothesis that tamoxifen administration *in utero* affects embryonic macrophage numbers in our models, we assessed testicular immune cell number at E13.5 and E18.5 after 4-OHT

administration at E12.5. We used immunofluorescence for CD45 and/or F4/80 in vehicle-only-injected versus 4-OHT-injected *Csf1r*-creER;*Rosa*-Tomato embryos pulsed at E12.5, as well as creER-negative;*Rosa*-Tomato embryos with 4-OHT injection. Quantitative stereology-based cell counts revealed that there was no effect on CD45+ cells at E13.5 or F4/80+CD45+ macrophages at E18.5; however, we found a statistically significant increase of F4/80-CD45+ cells (likely mostly monocytes) at E18.5 due to 4-OHT administration relative to vehicle injection. These findings suggest that only recruited CD45+ cells, and not tissue-resident macrophages, are increased in number long-term after tamoxifen administration. These new data are included in Supplementary Figures 1 and 2.

While the imaging data is beautiful and helps with localization of cell populations, inclusion of additional quantitative analysis approaches would help to strengthen the story because occasionally it is difficult to determine a conclusion from a narrow field of view. Image analysis or flow cytometric data would be welcomed. In addition, all bar plots should be changed to show replicates using symbols.

As requested, we have included a large number of additional quantitative analyses, including stereology-based counts of: *Kit*-creER-induced Tomato+ macrophages in E18.5 fetal brain (Supplementary Figure 6) and E12.5 fetal testes (Supplementary Figure 15); *Kit*-creER-induced Tomato+ macrophages in E18.5 and P30 liver (Supplementary Figure 6) and P7, P30, P60, and P120 gut (Supplementary Figure 8); *Csf1r*-creER-induced Tomato+ macrophages in E12.5 fetal testes (Supplementary Figure 15); cleaved-Caspase-3+ cells and CD45+ cells in E13.5 fetal testes (Supplementary Figure 1); F4/80+CD45+ cells and F4/80-CD45+ cells in E18.5 fetal testes (Supplementary Figure 2); SOX9+ Sertoli cells, ERG+ endothelial cells, and HSD3B1 or CYP17A1+ Leydig cells in E18.5 control and *Amh*-cre;*Rosa*-DTA fetal testes (Supplementary Figure 11); and HSD3B1+ Leydig cells, CD45+F4/80+ cells (macrophages) and CD45+F4/80- cells (monocytes) in E18.5 control and *Nr5a1*-cre; *Rosa*-NICD fetal testes (Supplementary Figure 12).

Furthermore, we have included new flow cytometric analyses to: (1) assess Tomato labeling of neutrophils, monocytes, and HSCs by tamoxifen-induced *Csf1r*-creER and *Cx3cr1*-creER activity in a more quantitative manner (as described in a previous response above) (Supplementary Figures 5 and 10); and (2) confirm our stereology-based counts of reduced immune cell numbers and immune cell populations in *Amh*-cre;*Rosa*-DTA fetal testes (Figure 5). Our flow cytometric analyses consistently verify and support our findings via immunofluorescence and qRT-PCR.

Finally, as requested, we have modified our bar graphs and plots to show all replicates.

Fig 5 A-L, can the ablation percentage of Sertoli cells and blood vessels be quantified? Does this result in changes in testosterone levels?

We performed rigorous quantification of SOX9+ Sertoli cells and ERG+ endothelial cells in E18.5 control and *Amh*-cre;*Rosa*-DTA fetal testes. We observed an approximately 80% reduction of Sertoli cells, while endothelial cell number was not significantly impacted. Additionally, to assess the effects of *Amh*-cre;*Rosa*-DTA on these cell types we performed qRT-PCR for Sertoli- and endothelial-specific genes, which also revealed a significant reduction in Sertoli cell gene expression with a negligible effect on endothelial cell gene expression. These new data are included in Supplementary Figure 11.

To assess effects on steroidogenesis, we used qRT-PCR to assess mRNA levels of *StAR*, *Cyp11a1*, *Hsd3b1*, and *Cyp17a1*, which encode steroidogenic enzymes. Furthermore, we also quantified Leydig cell number using stereology-based counts of HSD3B1+ and CYP17A1+ cells. We observed a ~60% reduction of CYP17A1+ cells and a ~35% reduction of HSD3B1+ cells. These new data are included in Supplementary Figure 11. These data indicate a likely reduction in androgen levels in *Amh-cre;Rosa-DTA* embryos. However, testosterone levels are ultimately determined by the activity of HSD17B3, which is expressed by Sertoli cells (and not Leydig cells) in the fetal testis (O'Shaughnessy et al., 2000 *Endocrinology* 141:2631-2637 and Shima et al., 2013 *Mol Endocrinol* 27:63-73); therefore, a significant ablation of Sertoli cells virtually guarantees that there will be a substantial reduction in testosterone levels in *Amh-cre;Rosa-DTA* embryos.

Do Sertoli cells express chemokines or maintenance factors to regulate monocyte recruitment (such as CCL2) or macrophage survival (such as Csf1) in the tissue?

We analyzed single-cell RNA-Seq data from a published dataset (Ademi et al., 2022 *Cell Reports* 39(11):110935) to determine whether E16.5 fetal Sertoli cells (i.e., around the onset of monocyte influx into the fetal testis) express *Csf1* or *Ccl2*. We found that Sertoli cells (clusters #2, 5, 11, and 12) do not appreciably express either of these genes; the only cell type that highly expresses either one of these genes is immune cells, which express *Ccl2*. However, these results are not surprising since CSF1R and CCR2 are not essential for testicular monocyte recruitment, as evidenced by the lack of any significant phenotype in monocytes after anti-CSF1R treatment in fetal stages (this study) and upon *Ccr2* mutation in adulthood (Lokka et al., 2020 *Nat Commun* 11(1):4375 and Wang et al., 2021 *PNAS* 118(1):e2013686117). We have provided these data for the reviewers here:

FIGURE REDACTED

To further address this point about chemokines in the Discussion, we provide a brief discussion of the potential role of CCR2 and another CSF ligand, CSF2, in testicular macrophage development.

The authors need to provide specifics regarding the number of experimental replicates performed for each assay and number of fully independent experiments. Representative plots are shown for a large number of panels, so it would be important to know the replication info, perhaps in the figure legends. I realize some of this information was present in the methods section, but it was unclear if this addressed for all of the experiments in the paper.

We agree with the reviewer that this is important information. In addition to including replicates for bar graphs and plots, we have described sample size in the figure legends.

Reviewer #2:

This paper demonstrates that adult testicular macrophages are derived from fetal hematopoietic stem cells (HSCs) using multiple lineage tracing experiments. Authors further claim that the Sertoli cells are essential for the 1) recruitment of monocytes in fetal testis and 2) differentiation of peritubular macrophages in postnatal testis. In addition, testicular macrophages' role in testis' prenatal development has been shown. The article is well written, and the diverse experiments support most conclusions. Although this is an elegant work, mouse adult testicular macrophages' ontogeny was largely deciphered in two recent studies (<https://doi.org/10.1073/pnas.2013686117>; <https://doi.org/10.1038/s41467-020-18206-0>). Like this study, both previous studies also concluded that adult testicular macrophages are derived mainly from fetal liver monocytes with minimal contribution from yolk sac progenitors and adult HSCs, which takes away some of the novelty of this study. However, the study remains exciting because the authors attempted to demonstrate the involvement of Sertoli cells in the recruitment of monocytes in fetal testis and the differentiation of peritubular macrophages in postnatal testis.

We thank the reviewer for their constructive comments. While we agree that previous publications have focused on the ontogeny of testicular macrophages, there has been some debate in the field and we felt that more definitive studies were needed. In this study we have performed an unprecedented and exhaustive analysis of this question using multiple mouse models. We believe that our results are extremely robust, clear-cut, and provide a conclusive answer. Not only have we clearly shown that fetal monocytes are the precursors for both adult testicular macrophage populations, but we have also defined a narrow fetal time window during which HSCs give rise to testicular immune cells. Furthermore, we have specifically addressed additional questions beyond testicular macrophage ontogeny, including the role of Sertoli cells, Leydig cells, and germ cells in immune cell recruitment; the role of postnatal Sertoli cells and germ cells in macrophage differentiation; and the function of specific adult macrophage populations in testosterone production by Leydig cells. The breadth of our findings in several areas of interest regarding testicular macrophages are a significant new source of information in the field, which we feel provide a great deal of novelty.

Specific comments that need to be addressed:

Major concern

1) In figure 2, the authors used the KitcreER fate mapping mouse model to demonstrate that adult testicular macrophages primarily originate from AGM-derived HSCs. Their observation is mainly based on Sheng et al., 2015. (<https://doi.org/10.1016/j.immuni.2015.07.016>) work. However, given late EMPs and AGM temporal promiscuity, KitcreER is not an ideal model to distinguish between late EMPs and AGM-derived HSCs. When pulsed with tamoxifen at E8.5, equal labeling of E10.5 EMPs and AGM multipotent progenitors (MPPs) has been reported (Sheng et al., 2015; C Stremmel et al., doi: 10.1038/s41467-017-02492-2). A similar observation was observed in this paper in Figure (2B, 20, 2N), and the authors concluded that "suggesting induction at E8.5 targets YS--derived EMPs and AGM derived

HSCs”.

It is well established that late EMP-derived fetal liver monocytes constitute the main precursor of adult macrophage populations, including Liver Kupffer cells. However, in supplementary Figure 4D, pulse labeling at E10.5, most of the F4/80+ Kupffer cells are robustly labeled with tomato (not mentioned in the main text). For the reasons stated above, I believe that the *KitcreER* fate-mapping model will not trace exclusively AGM-derived HSCs, but it may trace both late EMPs as well as AGM-derived HSCs (DOI:<https://doi.org/10.1016/j.immuni.2015.11.022>, doi: 10.1016/j.immuni.2016.02.024). Thereby, I believe that the data presented in this study do not support the claim that adult testicular macrophages are largely derived from AGM- HSCs.

We agree that the specificity of the *Kit*-creER is an important issue, but we respectfully disagree with some of the points brought up by the reviewer. First, we do agree that when pulsed with 4-OHT at E8.5, both YS-derived EMPs and AGM-derived MPPs are targeted by *Kit*-creER; as noted by the reviewer, this claim is supported by both Sheng et al., 2015 and by some of our data here. However, when *Kit*-creER is pulsed with 4-OHT at E10.5, our data suggests that only (or at least predominantly) AGM-derived HSCs are targeted. In Supplementary Figure 15, we can label E12.5 fetal testicular macrophages with *Csf1r*-creER and *Kit*-creER via an E8.5 tamoxifen pulse, which we believe corresponds to the early EMP population, since in Supplementary Figure 6, we can effectively label brain microglia with *Kit*-creER induced at E8.5. However, when we induce *Kit*-creER activity at E10.5, we observe virtually no labeling of E12.5 fetal testicular macrophages (Supplementary Figure 15), similar to E18.5 microglia (Supplementary Figure 6). If *Kit*-creER induced at E10.5 equally targeted late EMPs and AGM-derived HSCs, and if testicular macrophages arose from late EMPs, then we should see Tomato labeling of E12.5 testicular macrophages. Since we do not see any labeling of either macrophages or monocytes in E12.5 testes, the most logical conclusion is that the initial group of testicular macrophages arise from early EMPs (with little to no contribution from late EMPs), and they are eventually diluted out by AGM HSC-derived cells over the course of fetal development.

Furthermore, the timing of colonization of the fetal testis by monocytes (which eventually give rise to adult testicular macrophages) is not consistent with YS-derived progenitors, since Stremmel et al. showed that the peak of YS macrophage colonization of tissues peaks at around E10.5 and ends around E12.5 (Stremmel et al., 2018 *Nat Commun* 9(1):75); in contrast, the vast influx of testicular monocytes we observe begins only after E14.5, which is consistent with AGM HSC-derived progenitors. Additionally, Hoeffel et al., 2015 (*Immunity* 42:665–678) showed that monocytes started seeding the fetal liver at E13.5 and peaked as a percentage of immune cells at E14.5, while our data here show that monocytes are not detected in large numbers within the fetal testis until E16.5. Therefore, our data in sum strongly suggest that an E10.5 tamoxifen pulse using *Kit*-creER, with regard to the testis, predominantly targets AGM-derived HSCs and does not target YS-derived late EMPs and, subsequently, that monocytes migrating into the fetal testis are derived from AGM-derived HSCs. So while we agree with the reviewer that there is some potential “temporal promiscuity” between late EMPs and AGM MPPs, our data indicates that we can use *Kit*-creER to distinguish between these 2 populations and we can definitively identify the hematopoietic source of testicular macrophages.

Our *Flt3*-cre lineage-tracing data also provide evidence for an AGM HSC-derived progenitor for testicular macrophages, as nearly all adult testicular macrophages express Tomato in *Flt3*-cre;*Rosa*-Tomato mice, whereas a previous publication (Gomez Perdiguero et al., 2015 *Nature* 518:547–551) showed that only a small percentage of adult liver tissue-resident macrophages (Kupffer cells) express *Rosa*-YFP driven by

Flt3-cre. This result indicates that HSC-derived *Flt3*⁺ progenitors do not replace YS-derived macrophages in the liver (as well as in the brain, lung, and skin) to a high degree by early adulthood. Additionally, our analyses of P30 liver versus P30 testes in *Kit*-creER;*Rosa*-Tomato mice induced at E10.5 shows a differential pattern of targeting, in which only ~30% of P30 liver macrophages are Tomato-positive (Supplementary Figure 6), while 60-80% of P30 testicular macrophages are Tomato-positive (Figure 2). Overall, our data clearly show that macrophages of the testis are distinct in their origin as compared to the liver (and most other organs examined), in that AGM HSC-derived macrophages do appear to replace YS-derived macrophages in the testis almost completely by early adulthood.

We do acknowledge there are always potential technical limitations of lineage-tracing systems, but we feel that our data provides some definitive insights into this question. To reduce any confusion, we have deleted the discussion of the “fetal HSC hypothesis” and “EMP hypothesis” from the Discussion section.

2) *To demonstrate the role of Sertoli cells in the recruitment of fetal testicular monocytes, the authors depleted Sertoli cells by using the AMHCre-RosaDTX model. They denoted F4/80+ cells as macrophages and CD45+Iba1- as monocytes, which is factually incorrect. Iba1 is a marker of macrophages, and Iba1-negative cells should not be termed monocytes. It may be a combination of other immune cells, monocytes, neutrophils, and T cells.*

Although it is likely that monocytes comprise the vast majority of CD45+IBA1⁻ cells at this stage, we agree that describing CD45+IBA1⁻ cells broadly as monocytes is not entirely accurate. We have modified the text to acknowledge this fact and noted that this population at this stage may contain other immune cells such as neutrophils or other granulocytes.

*From E14.5 until birth, testis contains two macrophage populations: F80hi and F4/80int, with varying proportions (<https://www.nature.com/articles/s41467-020-18206-0>). F4/80Hi cells correspond to the yolk-sac-derived, and F4/80Int cells to the fetal liver monocyte-derived macrophages (doi: 10.1016/j.immuni.2015.03.011). If Sertoli cells recruit fetal monocyte to the testis, the proportion of F4/80Int cells should reduce in the AMHCre-RosaDTX mice compared to the control. Immunofluorescence analyses will not distinguish between F80hi and F4/80int cells. To examine conclusively whether Sertoli cells recruit fetal monocytes to the testis, authors should perform flow cytometry analysis using macrophage and monocyte-specific markers. This experiment should also be performed in the testis of *Dmrt1*—mouse.*

One caveat to interpreting the aforementioned results from the cited publication (Lokka et al., 2020 *Nat Commun* 11(1):4375) is that the authors showed strong Ly6C expression in virtually all F4/80-Int cells, suggesting that it is actually a monocyte population and not a macrophage population. Therefore, the F4/80-Int population reported by Lokka et al. is likely the round F4/80-dim or F4/80-negative cell population we visualize by immunofluorescence that is reduced in *Amh*-cre;*Rosa*-DTA fetal testes. However, we agree that flow cytometry would be a more sensitive way to distinguish between these populations, so we performed flow cytometry for immune cells in *Amh*-cre;*Rosa*-DTA gonads. Our data confirm that there is a specific reduction of CD45⁺ F4/80^{-lo} CD11b^{-hi} cells, which are likely monocytes, in Sertoli-depleted fetal testes, with no effect on F4/80^{-hi} cells. These new data are included in Figure 5.

As for *Dmrt1*-mutant mice, they do not experience loss of Sertoli identity or display a disrupted gonadal phenotype until after birth (Raymond et al., 2000 *Genes Dev* 14(20):2587-95; Fahrioglu et al., 2007 *Sex*

Dev 1(1):42-58; Kim et al., 2007 *Dev Biol* 307(2):314-27; and Matson et al., 2011 *Nature* 476(7358):101-4), so we think that *Dmrt1* mice are not a good model to address Sertoli-mediated monocyte recruitment during fetal stages.

3) *Ablation of the Sertoli cell population significantly reduced Leydig cell number* (DOI: 10.1210/en.2017-00196). *Because Leydig cells are necessary for the maintenance of testicular macrophages, aberrant change in the number of macrophages/monocytes in Sertoli cell depleted the Leydig cell number may indirectly influence testis. To rule out this possibility, the Leydig cell number should be determined by Stereology in the AMHCre-RosaDTX and Dmrt-/- mice.*

The reviewer brings up an excellent point. To address this question, we performed qRT-PCR for a number of Leydig cell genes and performed stereology-based cell counting of HSD3B1+ and CYP17A1+ Leydig cells in control and E18.5 *Amh-cre;Rosa-DTA* fetal testes (see comment above from Reviewer #1 regarding a similar question). We found there was a significant reduction in Leydig cell number, which is consistent with the aforementioned previous publication using this model that reported a reduction in Leydig cells in E16.5 and P0 *Amh-cre;Rosa-DTA* testes (Rebourcet et al., 2017 *Endocrinology* 158(9):2955-2969), as the reviewer rightly pointed out (although we found that CYP17A1+ cells were more severely reduced than HSD3B1+ cells). This new data is included in Supplementary Figure 11.

To follow up on this result, we tested the hypothesis that Leydig cells regulate immune cell recruitment into the fetal testis (independently of Sertoli cells), using *Nr5a1-cre; Rosa-NICD* embryos (also called *Rosa^{Notch}; Sfl-cre*), which exhibit a significant reduction in fetal Leydig cells while still retaining Sertoli cells (Tang et al., 2008 *Development* 135(22):3745-53). We found that, despite a significant and dramatic reduction of fetal Leydig cells in E18.5 *Nr5a1-cre; Rosa-NICD* fetal testes, there was no significant effect on immune cell numbers. This new data is included in Supplementary Figure 12.

As we noted in our response to the previous comment regarding *Dmrt1*-mutant mice, they do not experience loss of Sertoli identity or display a disrupted gonadal phenotype until after birth (Raymond et al., 2000 *Genes Dev* 14(20):2587-95; Fahrioglu et al., 2007 *Sex Dev* 1(1):42-58; Kim et al., 2007 *Dev Biol* 307(2):314-27; and Matson et al., 2011 *Nature* 476(7358):101-4), so we think that *Dmrt1* mutants are not a good model to address questions regarding Leydig cells or macrophages/monocytes in fetal stages.

Minor concern

Page 3, line 55-58; «... Furthermore, it has also been shown that a number of organs, such as the brain, maintain their tissue-resident macrophage populations (microglia) independently of bone-marrow-derived HSCs, while other organs, such as the gut, exhibit a heavy reliance on bone-marrow HSC-derived monocytes to maintain homeostasis of their tissue-resident macrophages». This sentence does not include information about fetal liver monocytes (FL-Mos) derived macrophages. Thus, the sentence should be modified accordingly.

We agree this information should be included. We have added a description about the contribution of fetal liver monocyte-derived macrophages in the Introduction.

Page 4, lines 63-68; «...The first wave, called primitive hematopoiesis, takes place in the embryonic yolk sac (YS) blood islands starting at embryonic day (E) day (E) 7.0 in mice and gives rise to primitive YS--

derived macrophages. Between E8.0 and E8.5, the second wave of hematopoiesis from YS hemogenic endothelium, sometimes referred to as the transient definitive wave, gives rise to a class of progenitors called erythromyeloid precursors (EMPs). After blood circulation starts at E8.5, EMPs migrate to the fetal liver and expand in number to differentiate into monocytes and other hematopoietic lineages».

The first wave of hematopoiesis at E7-7.5 give rise to early erythromyeloid precursors (EMPs) and the second wave of hematopoiesis at E8-8.5 give rise to late erythromyeloid precursors (EMPs). Authors should modify the above sentences for better clarity for the readers.

As requested, we have added a description of early versus late EMPs in this paragraph in the Introduction.

On Page 4, lines 64-75, important references are missing.

We have added several pertinent references in this paragraph of the Introduction.

Reviewer #3:

The study evaluates the origin of testicular macrophages using elegant cell tracking methods. This is a very elegant and well designed study that provides new information on the development of tissue macrophages, it challenge old concepts and provides new information. I suggest its publication

We thank the reviewer for their positive comments and enthusiasm for our study.

Minor comments: the discussion is too long and could be edited.

We agree. We have modified the Discussion to make it more concise, deleting much of the discussion of macrophage ontogeny and consolidating other parts of the text.

We are very grateful to the reviewers for their insightful and constructive comments that were invaluable in helping us strengthen our manuscript. Thank you very much for your consideration.

REVIEWERS' COMMENTS

Reviewer #1 (Remarks to the Author):

In the revised submission by Gu et. al., the authors address the origin of tissue resident testicular macrophages in development and their role in regulation of organ specific functions. Based on feedback provided during the initial review, the authors address many of the concerns raised by reviewers. Importantly, this includes additional control data, improved clarity in writing, and application of a new mouse model to test their overarching hypothesis. These new data pair well with prior fate-mapping and imaging data to strengthen their conclusions. Overall, the study was performed rigorously, is presented in a clear manner, and provides convincing data to support a novel conclusion. I have no additional concerns.

Reviewer #2 (Remarks to the Author):

I thank the authors for satisfactorily addressing almost all of my questions. The authors have effectively demonstrated the role of Sertoli cells in the requirement of testicular macrophages during neonatal life. However, I am still not fully convinced that the KitcreER fate-mapping model will unequivocally trace AGM-derived HSCs but not late EMPs-derived HSCs. The authors should discuss these limitations in the discussion section.

Reviewer comments are in italics and our responses are listed below in normal blue font.

Reviewer #1 (Remarks to the Author):

In the revised submission by Gu et. al., the authors address the origin of tissue resident testicular macrophages in development and their role in regulation of organ specific functions. Based on feedback provided during the initial review, the authors address many of the concerns raised by reviewers. Importantly, this includes additional control data, improved clarity in writing, and application of a new mouse model to test their overarching hypothesis. These new data pair well with prior fate-mapping and imaging data to strengthen their conclusions. Overall, the study was performed rigorously, is presented in a clear manner, and provides convincing data to support a novel conclusion. I have no additional concerns.

We thank the reviewer for their positive feedback about our revised manuscript.

Reviewer #2 (Remarks to the Author):

I thank the authors for satisfactorily addressing almost all of my questions. The authors have effectively demonstrated the role of Sertoli cells in the requirement of testicular macrophages during neonatal life. However, I am still not fully convinced that the KitcreER fate-mapping model will unequivocally trace AGM-derived HSCs but not late EMPs-derived HSCs. The authors should discuss these limitations in the discussion section.

We thank the reviewer for their constructive critiques. We respect and acknowledge the reviewer's view that there are potential technical limitations in *Kit*-creER fate-mapping experiments, so we have added a discussion of this model's limitations in the Discussion section. We also have softened our claims in the Discussion regarding the role of HSCs as hematopoietic progenitors for testicular macrophages.